# The dual molecular identity of vestibular kinocilia bridges structural and functional traits of primary and motile cilia

**Zhenhang Xu[1†], Amirrasoul Tavakoli[2\*†], Samadhi Kulasooriya[1], Huizhan Liu[1], Shu Tu[1], Celia Bloom[1], Yi Li[1], Tirone D Johnson[2], Jian Zuo[1], Litao Tao[1], Bechara Kachar[2\*], David Z He[1\*]**

[1]Department of Biomedical Sciences, Creighton University, Omaha, United States; [2]Laboratory of Cell Structure and Dynamics, National Institute on Deafness and Other Communication Disorders, National Institutes of Health, Bethesda, United States

**\*For correspondence:**
amir.tavakolitarghi@nih.gov (AT);
kacharb@nidcd.nih.gov (BK);
DavidHe@creighton.edu (DZH)

[†]These authors contributed equally to this work

**Competing interest:** The authors declare that no competing interests exist.

## eLife Assessment

Using single-cell transcriptomic data from mouse inner ear hair cells, the authors compare for the first time gene expression across the four recognized hair cell types in adults, generating information **fundamental** to understanding hair cell relationships between the ancient vestibular compartment and the more recent cochlea. Among observed differences, **compelling** evidence is provided for the expression in vestibular hair cells but not cochlear hair cells of certain ciliary motility-related genes, suggesting that the kinocilium of vestibular hair cells may function as an active force generator to increase sensitivity.

**Abstract** Vestibular hair cells (HCs) convert gravitational and head motion cues into neural signals through mechanotransduction, mediated by the hair bundle—a mechanically integrated organelle composed of stereocilia and a kinocilium. The kinocilium, a specialized form of primary cilium, remains incompletely defined in structure, molecular composition, and function. To elucidate its characteristics, we conducted single-cell RNA sequencing of adult vestibular and cochlear HCs, uncovering a selective enrichment of primary and motile cilia-associated genes in vestibular HCs, particularly those related to the axonemal repeat complex. This enrichment of orthologous axoneme-related genes was conserved in zebrafish and human vestibular HCs, indicating a shared molecular architecture. Immunostaining validated the expression of key motile cilia markers in vestibular kinocilia. Moreover, live imaging of bullfrog and mouse HCs from crista ampullaris revealed spontaneous kinociliary motion. Together, these findings define the kinocilium as a unique organelle with molecular features of primary and motile cilia and suggest its previously unknown role as an active, force-generating element within the hair bundle.

## Introduction

The mammalian inner ear contains auditory and vestibular end organs, which detect sound and motion signals, respectively. Hair cells (HCs), the sensory receptors found in both these structures, transduce mechanical stimuli in the form of sound or head movement into electrical signals (*Fettiplace, 2017*; *Gillespie and Müller, 2009*). The HCs of the auditory sensory epithelium in the cochlea are categorized as inner and outer HCs (IHCs and OHCs) (*Dallos, 1992*). The vestibular sensory epithelia in the two otolith organs (utricle and saccule) and three cristae associated with semicircular canals also

contain two types of HCs—types I and II—based on morphology, physiology, and innervation (*Eatock et al., 1998*; *Eatock and Songer, 2011*).

Although mechanotransduction is a shared feature of all HCs, mammalian cochlear and vestibular HCs differ in morphology and function. One key difference lies in the structure of the hair bundle, which harbors specialized machinery for mechanotransduction. The hair bundle of adult vestibular HCs is composed of actin-rich stereocilia connected via lateral links to a single microtubule-based kinocilium. Meanwhile, cochlear HCs lose their kinocilia during HC maturation (*Leibovici et al., 2005*). Kinocilia are highly conserved in non-mammalian vertebrates (*Leibovici et al., 2005*) and play a pivotal role in establishing hair bundle polarity and mediating Hedgehog and WNT signaling during HC development (*Moon et al., 2020*; *Shi et al., 2022*). Despite possessing features of motile cilia such as the canonical '9 + 2' microtubule arrangement, the kinocilium has long been regarded as a specialized primary cilium (*Kikuchi et al., 1989*; *Wang and Zhou, 2021*). While the kinocilium contributes to the bundle mechanics (*Baird, 1994*; *Kindt et al., 2012*; *Spoon and Grant, 2011*), the molecular basis and function of this unique organelle in adult vestibular HCs remain unknown.

In the current study, we utilized single-cell RNA-sequencing (scRNA-seq) to examine the transcriptomes of 1522 HCs isolated from cochlear and vestibular sensory epithelia of adult CBA/J mice. Comparisons of the mRNA profiles of the four HC types identified novel marker genes as well as shared and unique genes associated with mechanotransduction, ion channels, and pre- and post-synaptic structures. Notably, our analysis revealed a significant enrichment of genes related to primary and motile cilia in vestibular HCs, particularly those linked to the 96 nm axonemal repeat complex, a hallmark feature of motile cilia. Orthologous axoneme-related genes were also detected in zebrafish HCs and human vestibular HCs. We utilized transmission electron microscopy (TEM) to examine the ultrastructure of kinocilia and immunostaining to detect expression of key motile cilia proteins in vestibular HCs. We also used live imaging to examine kinocilia motion in bullfrog and mouse crista ampullaris. We modeled the atomic architecture of the 96 nm repeat, the core framework of the kinocilium axoneme. Together, these findings establish the kinocilium as a distinct organelle with molecular hallmarks of motile cilia and suggest it functions as an active, force-generating hair bundle component, influencing the mechanosensitivity of the kinocilium-bearing HCs across non-mammalian and mammalian species. In addition, our transcriptomic analysis provides new insight into the molecular mechanisms underlying phenotypical differences among the four different HC types in the adult mouse inner ear.

## Results

Solitary cells were isolated from the whole basilar membrane (together with the organ of Corti, *Figure 1A*) and vestibular end organs from 10-week-old CBA/J mice. Cells isolated from the cochlea include IHCs, OHCs, supporting cells (SCs), spiral ganglion neurons (SGNs), and other accessory cells. Some examples of individual IHCs and OHCs are shown in *Figure 1A*. Cells isolated from vestibular sensory epithelia include type I HCs, type II HCs, SCs, vestibular neurons, and other cell types. Vestibular HCs can be identified based on morphological features: Type I HCs are flask-shaped with a narrow neck, while type II HCs are cylindrical and short (*Burns and Stone, 2017*; *Eatock et al., 1998*; *Lysakowski and Goldberg, 1997*; *Pujol et al., 2014*; *Ricci et al., 1997*). Type II HCs and SCs also express *Sox2*/SOX2 (*Figure 1B*), which is a marker for these cell types in the vestibular sensory epithelium (*Jan et al., 2021*; *Wilkerson et al., 2021*). Some representative images of type I and II HCs isolated from maculae of utricle and saccule as well as from crista ampullaris are presented in *Figure 1C*.

To comprehensively assess the molecular profiles of inner ear HCs, we conducted scRNA-seq using cells isolated from the auditory and vestibular sensory epithelia. The within-tissue cellular diversity and identity in the cochlear and vestibular sensory epithelia were assessed by a t-distributed Stochastic Neighbor Embedding (t-SNE) analysis followed by clustering and annotation of cell types based on the expression of known marker genes (*Figure 1D, E*; *Burns et al., 2015*; *Jan et al., 2021*; *McInturff et al., 2018*; *Ranum et al., 2019*; *Wilkerson et al., 2021*; *Xu et al., 2022*). Upon cluster annotation and normalization across biological repeats, HC types were separated for the downstream analysis by their known marker genes shown in feature plots (*Figure 1F*) and dot plots (*Figure 1G*). A total of 1522 individual cells were identified as HCs, including 131 IHCs, 668 OHCs, 588 type I HCs, and 135 type II HCs for our downstream analysis. We aggregated the number of reads for each gene across

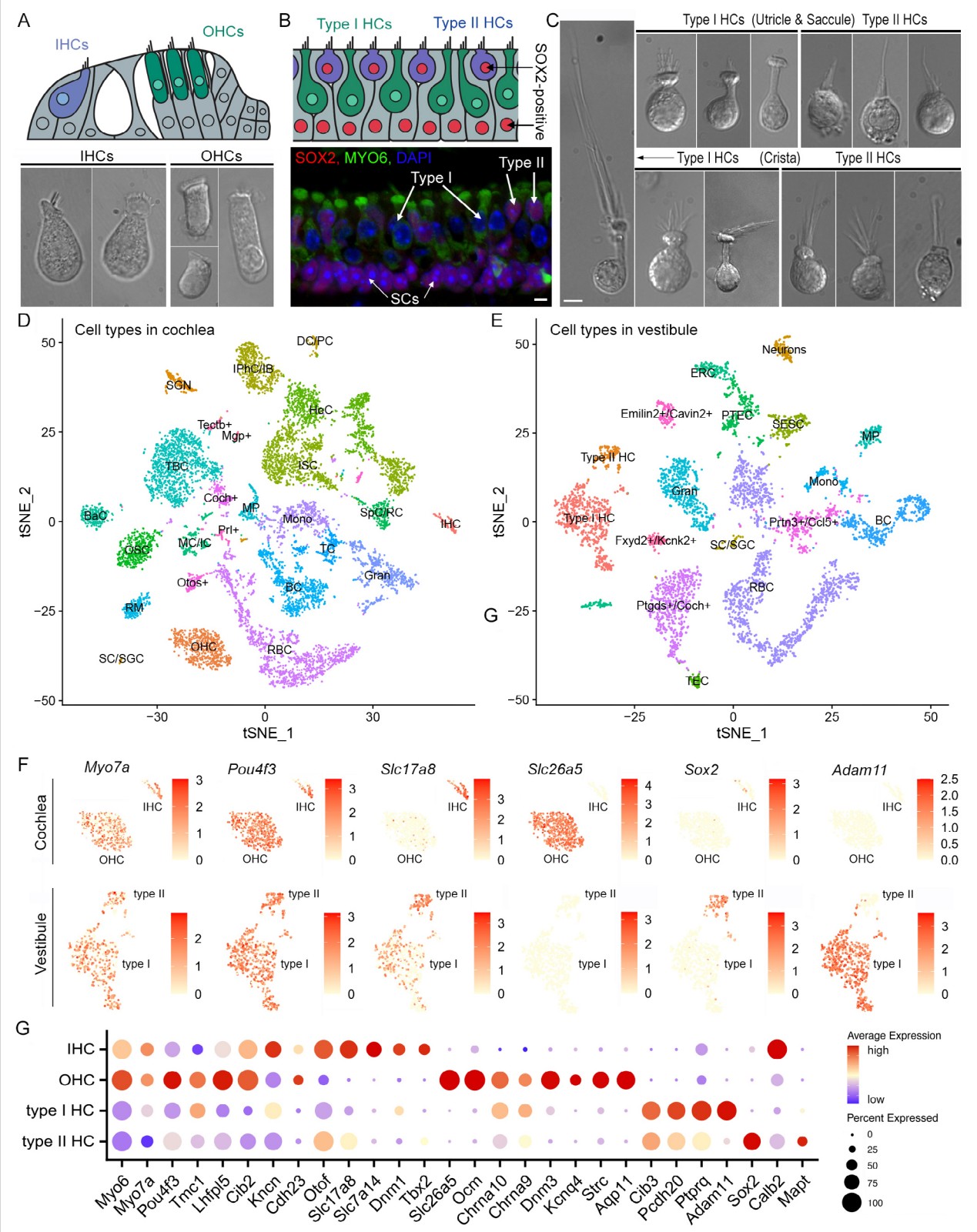

**Figure 1.** Single-cell transcriptional atlas of cochlear and vestibular cells. (**A**) Schematic drawing of the organ of Corti (top panel) and representative images of IHCs and OHCs from adult mouse cochleae. (**B**) Schematic drawing of the utricle (top panel) and confocal images of the utricle prepared from an adult mouse. HCs are stained with MYO6, SOX2, and DAPI. SOX2-positive cells include type II HCs and SCs underneath HCs. (**C**) Representative images of type I and II HCs from utricular and saccular maculae as well as crista ampullaris from adult mice. (**D, E**) tSNE plots of distinct cell types

*Figure 1 continued*

detected in the adult CBA mouse cochlea (**D**) and utricle, saccule, and crista. Different cell types are color-coded. (**F**) Feature plots of the expression six marker genes in different HC populations. (**G**) Dot plot heatmap of average expression and cellular detection rate of 28 representative marker genes in different HC types in cochleae and vestibular end organs. Abbreviations: IHC (inner HC); OHC (outer HC); SGN (spiral ganglion neuron); SC (Schwann cell)/SGC (satellite glial cell); DC (Deiters' cell)/PC (pillar cell); IPhC (inner phalangeal cell)/IB (inner border cell); ISC (inner sulcus cell); HeC (Hensen's cell); SpC (spindle cell)/RC (root cell); MC (marginal cell)/IC (intermediate cell); BaC (basal cell); MP (macrophage); RM (Reissner's membrane); BC (B cell); TC (T cell); Gran (granulocyte); Mono (monocyte); RBC (red blood cell); OSC (outer sulcus cell); TBC (tympanic border cell). Type I HC (type I HC); Type II HC (type II HC); SESC (sensory epithelial SC); TEC (transitional epithelial cell); PTEC (peripheral transitional epithelial cell); ERC (epithelial roof cell). Neurons (vestibular neurons). For those cells whose definite identities cannot be annotated, the top expressed genes were used for identification and annotation. These cells include Tectb$^+$, Mgp$^+$, Coch$^+$, Prl$^+$, Otos$^+$, Emilin2$^+$/Cavin2$^+$, Fxyd2$^+$/Kcnk2$^+$, Prtn3$^+$/Ccl5$^+$, and Ptgds$^+$/Coch$^+$.

all single cells to generate pseudo-bulk expression profiles for different HC types (*Supplementary file 1*).

## Similarities among HC types

We utilized the pseudo-bulk and single-cell gene expression profiles to compare the four HC types. Since adult IHC and OHC transcriptomes have been compared extensively in the past (*Li et al., 2018*; *Ranum et al., 2019*), we focused our analyses on the differences between cochlear and vestibular HCs, as well as between type I and II HCs. Principal component analysis (PCA) was used to denote variance across the four types of HCs. The first principal component (PC1) shows that the most dramatic differences are tissue-based—between cochlear and vestibular HCs (*Figure 2A*). The distribution across PC2 indicates that the similarity between type I and II HCs is greater than the similarity between IHCs and OHCs, suggesting more homogeneity among vestibular HCs. Gene expression profiles of single HCs were also used to analyze similarities among the four different HC types (*Figure 2B*). Despite heterogeneity found among individual HCs of each type, three-dimensional PCA visualization of single cells draws similar conclusions to the pseudo-bulk PCA, suggesting higher transcriptomic similarity between vestibular HCs in contrast to cochlear HCs.

## Differentially expressed genes in HC types

Next, we examined the number of genes that are shared and unique among the four types of HCs based on pseudo-bulk expression profiles (*Figure 2C*). Although approximately 71% of the detected genes are shared among all four types of HCs, the smaller proportion of unique genes in each HC population may underlie their specific biological identities. We performed pairwise differentially expressed gene (DEG) analyses between within-tissue HC types, as well as between cochlear and vestibular HCs (*Figure 2D*). DEGs were defined as those with an expression level above 0 and a minimum of twofold change (log$_2$ ≥1) between the two cell populations with statistical significance of p ≤ 0.01.

DEGs between adult IHCs and OHCs have been analyzed before using cell type-specific microarray and bulk RNA-seq techniques (*Li et al., 2018*; *Liu et al., 2014*). Here, our comparison revealed differential enrichment of 154 and 123 genes in IHCs and OHCs, respectively (*Figure 2D*). We show expression of previously characterized DEGs in IHCs (e.g., *Otof* and *Slc17a8*) and OHCs (e.g., *Ocm*, *Slc26a5*, and *Chrna10*) as well as genes whose functions have not yet been characterized, including *Atp2a3*, *Calb2*, *Dnajc5b*, *Ripor3*, and *Scd1* in IHCs, and *Aqp11*, *Dnm3*, and *Sh3gl3* in OHCs. Our DEG analysis comparing IHCs and OHCs is consistent with the previous studies (*Li et al., 2018*; *Liu et al., 2014*).

We next compared DEGs between type I and II HCs (*Figure 2D*). Although some new marker genes of adult type I and II HCs from mouse utricles have been identified (*McInturff et al., 2018*), differential gene expression analysis between type I and II HCs has not been conducted extensively. Our analysis revealed enrichment of 35 DEGs in type I HCs and 59 DEGs in type II HCs. Except for a few genes such as *Otog* and *Bmp2*, the roles of these DEGs in vestibular HCs have not been examined.

Grouping the tissue-specific subtypes together, we calculated DEGs between cochlear and vestibular HCs (*Figure 2D*). This comparison identified 674 DEGs enriched in cochlear HCs and 887 DEGs enriched in vestibular HCs. We note that many vestibular DEGs (such as *Adam11*, *Car12*, *Nrxn3*, *Lpgat1*, *Spp1*, *Pcdh20*, and *Cfap126*) are related to the secretion of extracellular matrix protein, cell–matrix interactions, cell adhesion, mineralized matrix, calcification, acid–base balance, and cilia. Meanwhile, many of the genes enriched in cochlear HCs (such as *Slc25a5*, *Strip2*, *C1ql1*, *Rorb*, and *Ocm*) are

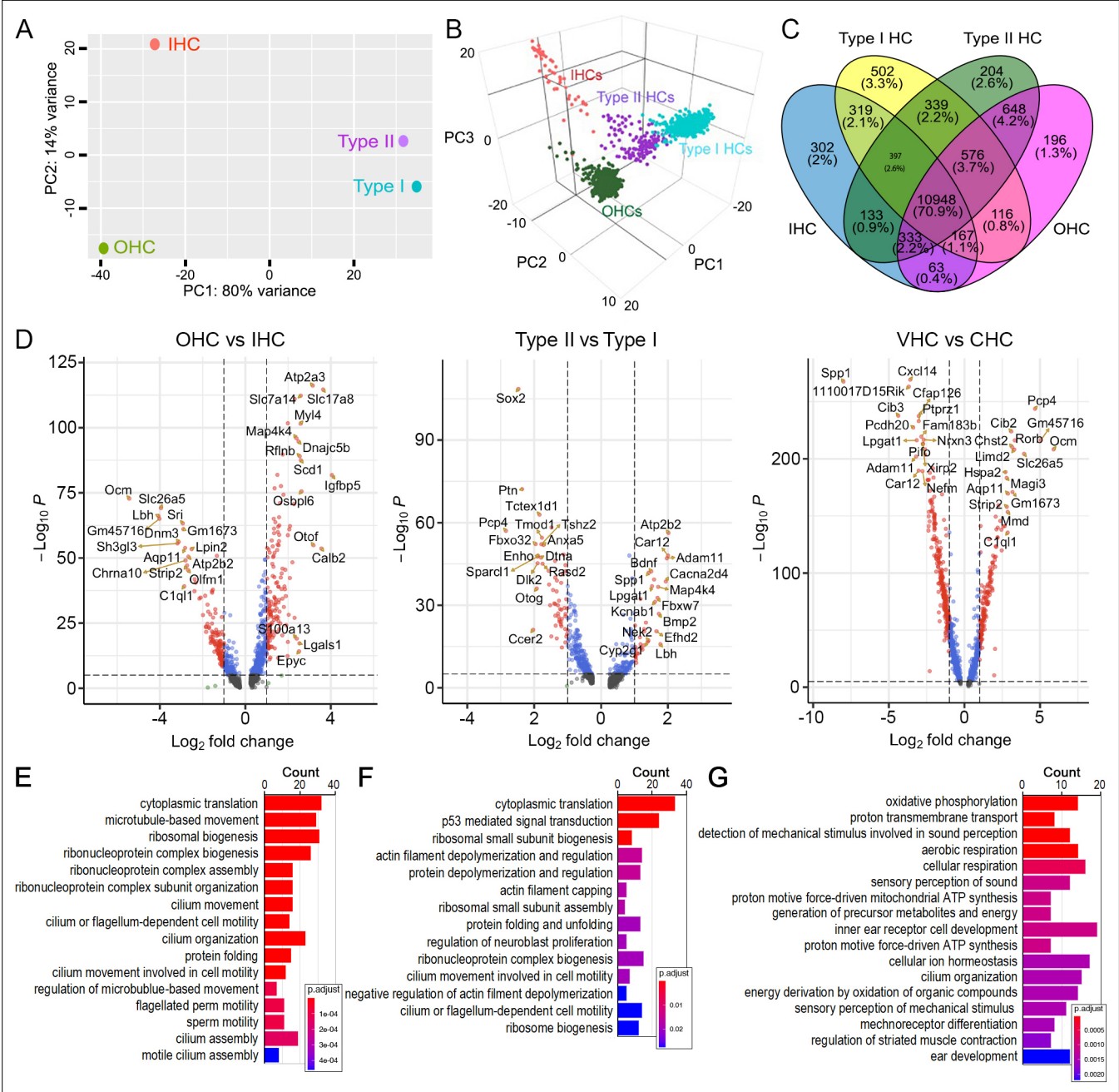

**Figure 2.** Similarity and difference among different HC types and biological processes enriched in cochlear and vestibular HCs. (**A**) Principal component analysis (PCA) plot showing similarity based on pseudo bulk RNA-seq data from four HC types. (**B**) PCA plot showing similarity based on individual HC gene expression among the four HC types. (**C**) Venn diagram depicting the number of expressed genes (RPKM >0) in four HC types. (**D**) Volcano plot showing differentially expressed genes between different HC types. Red dots indicate differentially expressed genes with p-value <10$^e$−5 and log$_2$ fold change >1. Only the top 20 differentially expressed genes are labeled. (**E**) Biological processes enriched in vestibular HCs compared to cochlear HCs. Biological processes related to motile cilia are enriched. (**F**) Biological processes enriched in type I HCs compared to type II HCs. (**G**) Biological processes enriched in type II HCs compared to type I HCs.

related to the extensively studied unique structure and function of cochlear HCs. It is interesting that *Cib2* is differentially expressed in cochlear HCs while *Cib3* is enriched in vestibular HCs. *Cib2* and *Cib3* act as an auxiliary subunit of the sensory mechanoelectrical transduction channel in HCs (*Giese et al., 2017*; *Riazuddin et al., 2012*; *Wang et al., 2023*), and mutations of these two genes are associated with deafness and Usher syndrome 1J (*Riazuddin et al., 2012*).

To examine the functional relevance of the calculated DEGs, we conducted overrepresentation analysis (ORA). Since the molecular properties of vestibular HCs are less known compared to cochlear HCs, we looked more closely at the biological processes enriched in vestibular HCs compared to cochlear HCs, which included gene ontology (GO) terms related to cilium organization and microtubule-based cilia motility (highlighted by red asterisks in *Figure 2E*). We also conducted ORA between type I and II HCs. Enriched processes in type I HCs included those related to cytoplasmic translation, p53-mediated signal transduction, actin filament depolymerization and regulation, and cilium or flagellum-dependent cell motility (*Figure 2F*). Terms enriched in type II HCs are related to oxidative phosphorylation, proton transmembrane transport, aerobic and cellular respiration, detection of mechanical stimulus involved in sound perception, and mechanoreceptor differentiation (*Figure 2G*).

## Marker genes for different HC types and genes related to HC specialization

Previous studies have characterized some marker genes in different HC types. We assessed the expression of previously characterized and newly identified genes related to HC structure and function across all four HC types. Many genes are expressed in all four HC types (*Figure 3A*). Among these are well-characterized genes such as *Cib2*, *Otof*, *Kcna10*, *Ptprq*, *Tmc1*, and *Espn*, while other genes such as *Tjap1*, which encodes a tight junction-associated protein, are less known. We also noted genes that were enriched in a tissue-specific manner. For example, *Cdh23*, *Rorb*, and *Osbp2* are more highly expressed in cochlear HCs than in vestibular HCs, while genes such as *Ldhb*, *Fbxo32*, *and Gsn* are enriched in vestibular HCs but only weakly expressed in cochlear HCs. Several genes expressed in vestibular HCs with no expression in cochlear HCs (such as *Cfap43*, *Cfap126*, *Cib3*, *Cxcl14*, *Pcdh20*, *Pifo*, *Slc9a3r2*, and *Tmc2*) could potentially be used as vestibular HC markers (*Figure 3A*). Moreover, *Adam11*, *Cacna2d4*, *Car12*, and *Shank2* are found to be only expressed in type I HCs, while *Ccer2*, *Cfap45*, *Dlk2*, and *Rprm* are only expressed in type II HCs. Our analysis also revealed new marker genes for IHCs (*Atp2a3*, *Rims2*, and *Ripor3*) and OHCs (*Aqp11*, *Mmd*, and *Sh3gl3*). *Cfap43*, *Cfap44*, *Cfap45*, *Cfap126*, *Kif3*, *Mlf1*, and *Pifo*, enriched in vestibular HCs, are all associated with kinocilium structure and function. We utilized single-molecule fluorescent in situ hybridization (smFISH) and immunostaining techniques to validate the expression of 12 genes in HCs (*Figure 3B, C*). The expression patterns of these genes and proteins were highly consistent with our observations from scRNA-seq analysis. Overall, our results identified novel marker genes and previously unidentified expression patterns among the four HC types.

Next, we focused our analysis on evaluating the genes associated with HC function. We compared the expression of 208 genes related to HC structure and function, including stereocilia and apparatus for mechanotransduction, ion channels, and synapses (*Figure 4A*). Among the genes associated with stereocilia and mechanotransduction apparatus (*Shin et al., 2013*), *Calm1*, *Calm2*, *Eps8*, *Espn*, *Fbxo2*, *Dynll2*, *Ush1c*, *Ywhae*, *Tmc1*, *Tmie*, *Cdh23*, *Pcdh15*, *Ank1*, *Ank3*, and *Lhfpl5* were highly expressed in all four HC populations. Others were highly expressed in only one or two HC populations. *Atp2a3*, *Calb2*, and *Dpysl2* were highly expressed in IHCs, while *Lmo7*, *Ocm*, and *Strc* were highly expressed in OHCs. Moreover, expression of *Dpysl2*, *Cdh23*, and *Cib2* was higher in cochlear HCs than in vestibular HCs, while *Xirp2*, *Pls1*, *Slc9a3r2*, and *Tubb4b* were more highly expressed in vestibular HCs than in cochlear HCs. Some genes were uniquely expressed in either cochlear or vestibular HCs. For example, *Cib3* and *Tmc2* are expressed in vestibular HCs but not in cochlear HCs.

All HCs possess ion channels. Our analysis detected several genes related to stretch-activated ion channels, such as *Trpc1* and *Trpm4*. Genes for Cl⁻ and Na⁺ channels were expressed. For Ca²⁺ and K⁺ channels, *Cacna1d* and *Kcna10* were expressed in all four HC types with varying levels of expression. In cochlear HCs, *Cacna1d*, *Kcna10*, *Kcnab1*, *Kcnj16*, and *Kcnma1* were expressed in IHCs, whereas *Cacna1d*, *Kcna10*, *Kcnk1*, *Kcnma1*, *Kcnn2*, and *Kcnq4* were expressed in OHCs. In the vestibular HCs, *Cacna2d4*, *Kcna10*, *Kcnab1*, and *Kcnma1* showed relatively high expression in type I HCs, whereas

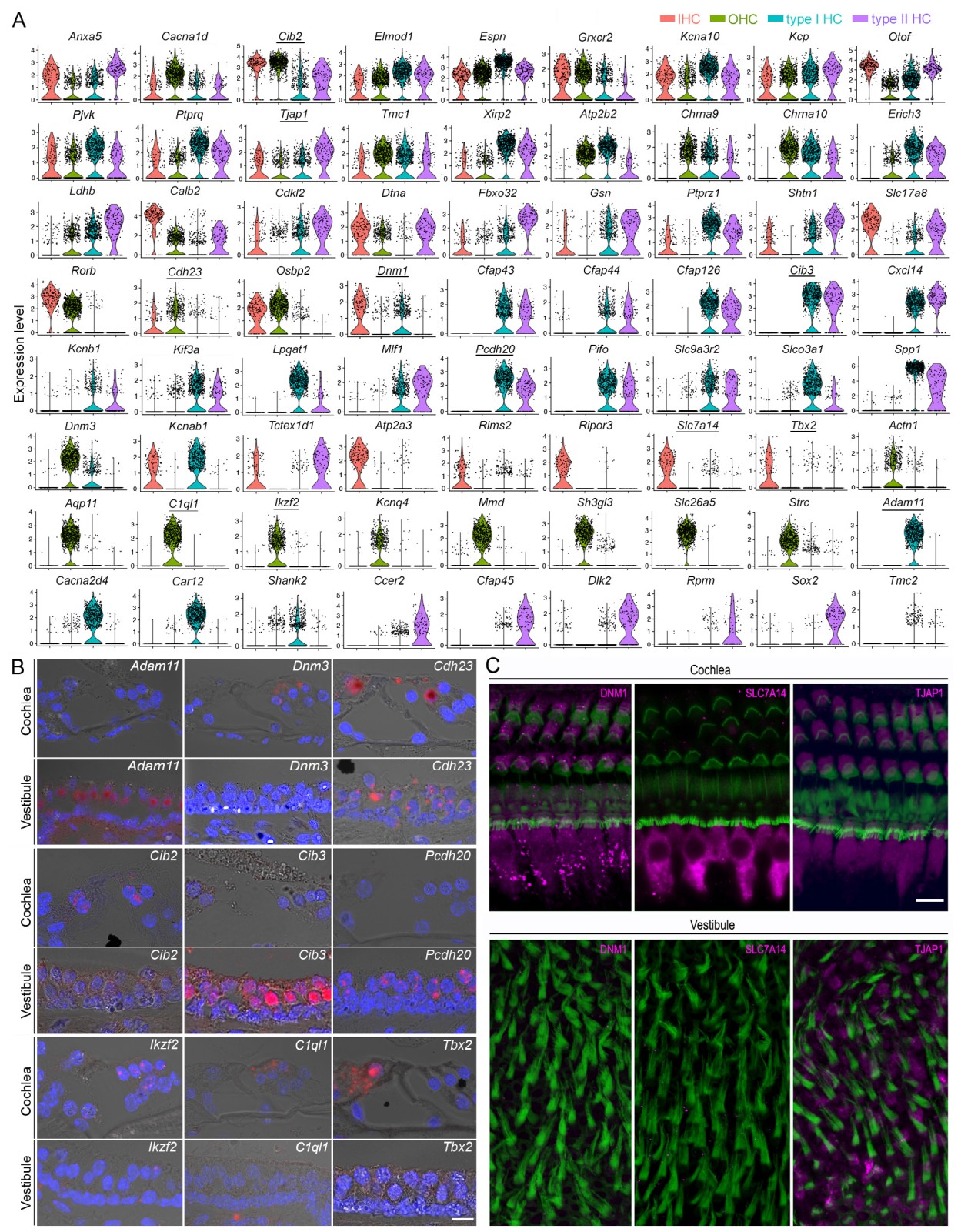

**Figure 3.** Shared and unique genes expressed in cochlear and vestibular HCs. (**A**) Violin plots of the expression of 72 genes in four different HC types. (**B**) Validation of differential expressions of nine genes (with underline in **A**) in cochlear and vestibular HCs in thin section. Bar: 10 μm for all images in B. (**C**) Confocal images of expression of DNM1, SLC7A14, and TJAP1 in cochlear and vestibular HCs. Bar: 10 μm for all images in C.

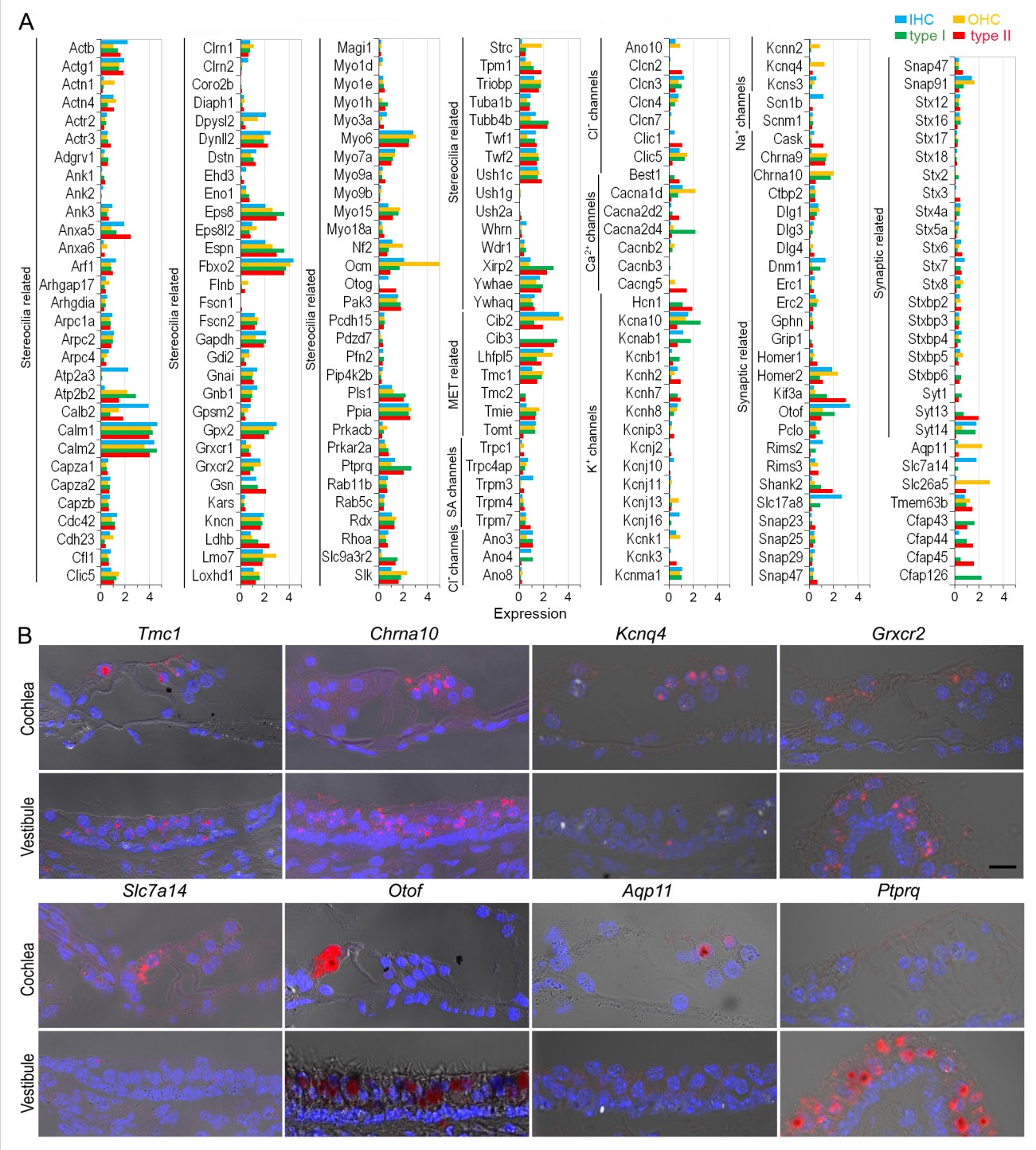

**Figure 4.** Expression of genes related to HC specialization. (**A**) Heatmap showing expression of genes related to stereocilia bundles, mechanotransduction, ion channels, and synaptic structure. (**B**) Validation of gene expression using single-molecule fluorescent in situ hybridization (smFISH). Bar: 10 µm.

type II HCs indicated a relatively high expression of *Cacna2d4*, *Cacng5*, *Kcna10*, *Kcnb1*, *Kcnh2*, and *Kcnh7*. *Best1*, *Clic*, *Hcn1*, and *Kcnh7* were only expressed in vestibular HCs.

Next, we examined the genes related to synapses (*Figure 4A*). Our results indicated expression of *Ctpb2*, *Dlg1*, *Homer2*, *Otof*, *Pclo*, and *Snap91* in all HCs at varying levels. We observed a relatively higher expression of *Dnm1*, *Otof*, *Rims2*, *Slc17a8*, *Snap91*, and *Stx7* in IHCs, while *Dlg1*, *Dnm3*,

*Snap9*, *Chrna9*, and *Chrna10* showed higher expression in OHCs. Some of the highly expressed genes in type I HCs include *Dnm1*, *Kif3a*, *Pclo*, *Shank2*, *Slc17a8*, *Syt13*, *Syt14*, *Chrna9*, and *Chrna10*, whereas type II HCs showed relatively higher expression of *Kif3a*, *Otof*, *Shank2*, *Stx7*, and *Syt13*. We used smFISH to validate the expression of 8 additional genes across the four HC types. The expression patterns shown in *Figure 4B* are consistent with our analysis (*Figure 4A*).

## Gene signatures of primary cilia in cochlear and vestibular HCs

A key morphological feature in the hair bundle of vestibular HCs is the presence of kinocilium, which has been regarded as a type of specialized primary cilia. Since our analysis revealed an enrichment of cilium-related GO terms (*Figure 2F*) and axonemal genes such as *Cfap43*, *Cfap44*, *Cfap45*, *Cfap126*, *Kif3*, *Plf1*, and *Tubb4b* (*Figure 3A*) in vestibular HCs, we sought to investigate the composition and molecular nature of the kinocilium (*Figure 5A*). Proteomics-based approaches have contributed to the development of cilia-associated protein databases. We utilized several well-established databases, including CiliaCarta (*van Dam et al., 2019*), the SYSCILIA gold standard (SCGSv2) (*Vasquez et al., 2021*), and CilioGenics (*Pir et al., 2024*) to compile a list of ~1000 cilia-related genes. We noted a significant overlap of these genes with our HC transcriptomic profiles (*Figure 5B*). GO analysis of the overlapping genes revealed enrichment of cellular component and biological process terms primarily related to cilia organization, assembly, maintenance, intracellular transport, and microtubule dynamics, particularly associated with motile cilia (*Figure 5—figure supplement 1*). Additionally, molecular function analysis highlights associations with motor activity, dynein chain binding, and BBSome binding (*Figure 5—figure supplement 1*). BBSome (Bardet–Biedl syndrome) is an octameric protein complex crucial for regulating transport in primary cilia (*Tian et al., 2023*).

The primary cilium is a sensory organelle that responds to and transmits external signals to the interior of the cell. Structurally, primary cilia are characterized by the presence of nine microtubule doublets encircling the shaft, which transition into a disorganized structure distally (*Figure 5C*). We first examined the expression of primary cilia-related genes in vestibular and cochlear HCs. Among approximately 420 primary cilia-related genes/proteins (*van Dam et al., 2019*; *Pir et al., 2024*; *Vasquez et al., 2021*), we detected the expression of 410 genes in at least one type of HC. *Figure 5D* shows the top 50 abundantly expressed primary cilia-related genes in type II HCs compared to the other three HC types. Most of these genes were detected in all four HC types except a few genes, such as *Mlf1*, *Ttc21a*, and *Tmem218*, which were weakly or not expressed in cochlear HCs.

Primary cilia are enriched in receptors and effectors for key pathways, including GPCR, cAMP, $Ca^{2+}$, RTK, TGF-β, MAPK, TOR, BMP, Wnt, Notch, and Rho signaling (*Anvarian et al., 2019*), localized to the ciliary shaft, transition zone, and BBSome (*Hansen et al., 2025*). We analyzed the expression of these genes in HCs and found higher expression levels of these pathway-related genes in vestibular HCs than in cochlear HCs (*Figure 5—figure supplement 2*).

Intraflagellar transport (IFT) involves anterograde and retrograde transport of molecules along the axoneme of cilia, facilitating the transport of components between the ciliary base and tip (*Ma et al., 2023*). IFT is essential for the proper assembly and maintenance of both primary and motile cilia. Thus, we assessed the expression of IFT-associated genes in HCs. Most of the genes are enriched in vestibular HCs compared to cochlear HCs (*Figure 5D*). Immunostaining confirmed the expression of IFT172 and CLUAP1 in the kinocilia of vestibular HCs (*Figure 5E*).

## Gene signatures of motile cilia in vestibular HCs

Curiously, our GO analysis returned many terms related to cilium motility. Motile cilia are highly conserved organelles across different organisms and tissues, although they exhibit organism- and tissue-specific adaptations (*Leung et al., 2025*). Recent advances using proteomics of isolated motile cilia from various ciliated tissues have enabled the profiling of genes associated with motile cilia. To explore whether the kinocilium possesses a molecular composition characteristic of motile cilia, we compared our HC transcriptomes with multi-tissue proteomics datasets derived from different organisms, including human, bovine, porcine, and murine, as well as diverse motile ciliary tissues such as sperm, oviduct, ventricle, and trachea (*Leung et al., 2025*; *Figure 5—figure supplement 3*). Our analysis revealed a significant overlap, particularly among axonemal components of motile cilia, with strong enrichment in vestibular HCs compared to cochlear HCs. The axoneme of motile cilia and flagella is a cylindrical structure harboring a canonical '9 + 2' arrangement, where nine doublet

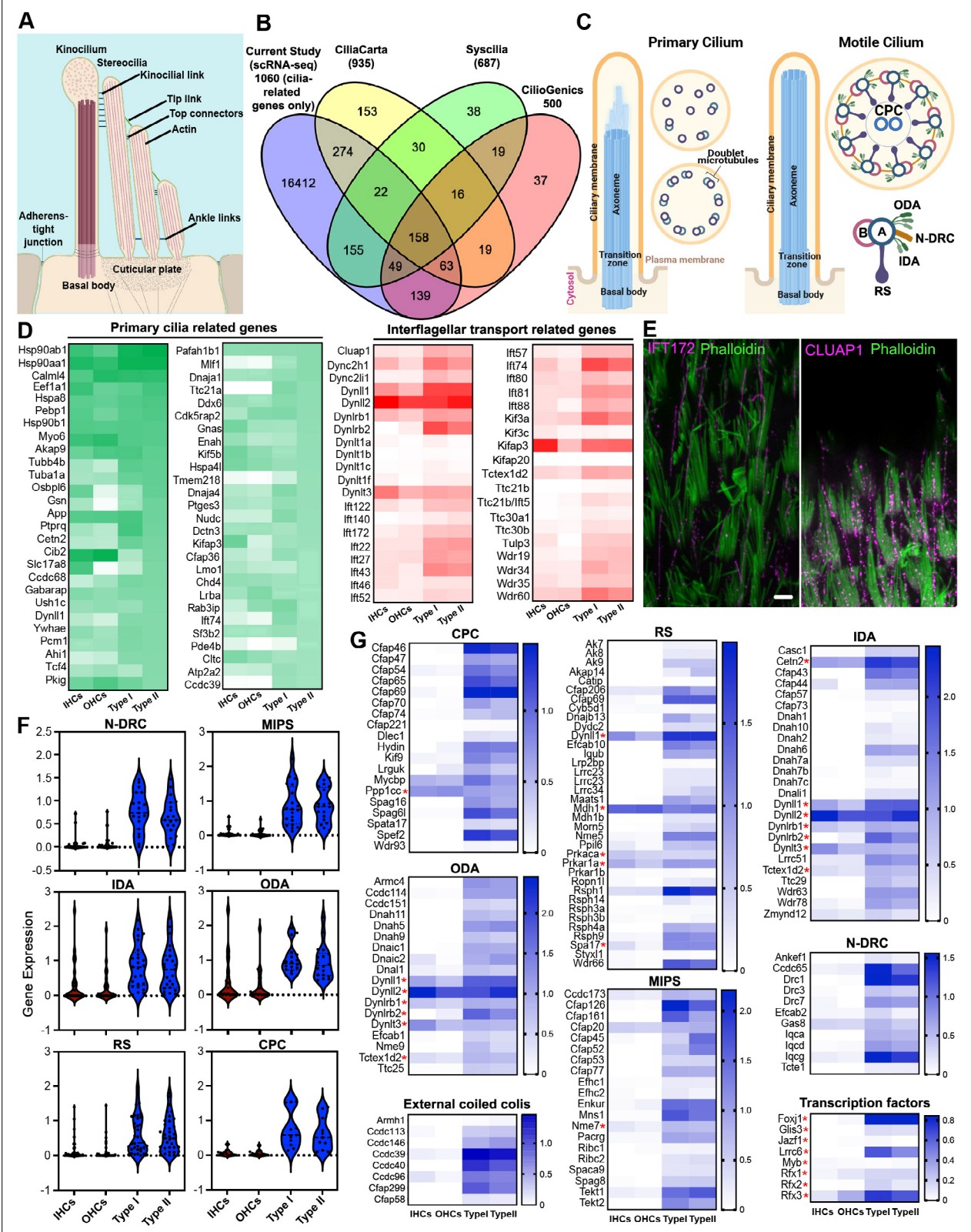

**Figure 5.** Cilia-related genes detected in cochlear and vestibular HCs. (**A**) Schematic illustration of HC hair bundle (adapted from *Schwander et al., 2010*). (**B**) Venn diagram of the number of genes in each database and the cilia-related genes detected in HC transcriptomes. (**C**) Schematic illustration of primary and motile cilia, highlighting 9 + 0 or 9 + 2 arrangement of microtubules for primary and motile cilia, respectively: radial spokes (RS), central pair complex (CPC), nexin–dynein regulatory complex (N-DRC), microtubule inner proteins (MIPs), inner and outer dynein arms (IDA and ODA). (**D**)

*Figure 5 continued on next page*

*Figure 5 continued*

Expression of top 50 cilia-related genes and genes related to IFT in the four types of HCs. (**E**) Immunostaining of IFT172 and CLUAP1 expression in vestibular kinocilia. Bar: 5 μm. (**F**) Violin plots showing aggregated expression of genes associated with 96 nm repeat. Expression value of these genes is based on *Supplementary file 1*. (**G**) Heatmaps of comparison of gene expressions related to motile cilia machinery in cochlear and vestibular HCs. Red asterisks indicate the genes whose encoded proteins are expressed in both cilia and cytoplasm or are multifunctional.

The online version of this article includes the following figure supplement(s) for figure 5:

**Figure supplement 1.** Gene ontology (GO) analysis of shared genes in the current study.

**Figure supplement 2.** Primary cilia-related genes in four HC subtypes.

**Figure supplement 3.** Comparison of vestibular HC single-cell RNA-sequencing (scRNA-seq) data with proteomics datasets from multiple tissues and species containing motile cilia.

**Figure supplement 4.** Enhanced accessibility of motile cilia-associated gene loci in adult vestibular HCs based on published ATAC-seq data (*Jen et al., 2019*).

microtubules (DMTs) surround two microtubule singlets in the center (*Figure 5C*). Altogether, the axoneme machinery consists of nine DMTs, two rows of inner and outer dynein arms (IDAs and ODAs), nexin–dynein regulatory complex (N-DRC), two singlet central pair complexes (CPCs), three radial spokes (RSs), microtubule inner proteins (MIPs), and external coiled-coil regions. The motility unit is arranged in a 96-nm repeat module along the CPC. Recent cryo-electron microscopy (cryo-EM) and cryo-electron tomography (cryo-ET) studies have provided a more comprehensive identification of this module repeat (*Chen et al., 2023*; *Leung et al., 2025*; *Walton et al., 2023*). To understand the molecular composition and function of kinocilia, we focused our analysis on the expression of genes associated with the structure of motile cilia. First, we assessed the expression of gene sets related to the motile cilium and each of its structural components. We noted a robust expression of motile cilia-related gene signatures in vestibular HCs, while cochlear HCs expressed little to none (*Figure 5F*). Next, we further assessed the expression of the key genes related to motile cilia machinery (*Figure 5G*), including the 96 nm module and CPC, based on the current known localization from biochemistry and proteomics.

We examined genes related to the 96 nm axonemal repeat of mammalian epithelial cilia. This structural unit contains proteins encoded by 128 genes (*Gui et al., 2021*). We found that 112 of these genes were expressed in adult vestibular HCs (*Supplementary file 2*). Genes encoding axonemal dynein (IDAs and ODAs), such as *Dnah5* and *Dnah6*, as well as RS components (*Wdr66*, *Cfap206*, *Cfap61*, and *Iqub*), N-DRC components (*Drc1* and *Iqca*), and MIPs (*Cfap126* and *Wdr63*) were predominantly expressed in vestibular HCs with little to no expression in cochlear HCs (*Figure 5G*). Axonemal CCDC39 and CCDC40, which form external coiled-coil regions, are the molecular rulers that organize the axonemal structure in the 96 nm repeating interactome and are required for the assembly of IDAs and N-DRC for ciliary motility (*Becker-Heck et al., 2011*; *Brody et al., 2025*; *Merveille et al., 2011*; *Oda et al., 2014*). Our results indicate a high expression of *Ccdc39* and *Ccdc40* in vestibular HCs, whereas little to no expression was observed in cochlear HCs. We should point out that unlike axonemal dynein proteins, which are uniquely required for cilia motility, the encoded proteins of several genes in *Figure 5G* (marked by red asterisks) are also expressed in cytoplasm and/or are multifunctional.

Next, we examined the expression of genes encoding transcription factors that are known key regulators of ciliome gene activation, including RFX and FOXJ1 transcription factor families. RFX controls genes in both motile and non-motile cilia, while FOXJ1 specifically governs motile cilia formation (*Choksi et al., 2014*). Our data show moderate expression of *Foxj1* in vestibular HCs and weak expression in cochlear HCs (*Figure 5G*). Other genes that regulate motile cilia formation, including *Lrrc6*, were also expressed at high to moderate levels in vestibular HCs compared to cochlear HCs. We also found a strong enrichment of transcriptional targets associated with vestibular HCs, particularly those involved in motile cilia programming and maintenance. The expression of these transcription factors may reflect their importance in the maintenance of kinocilia in adult vestibular HCs. Furthermore, analysis of a published ATAC-seq dataset (*Jen et al., 2019*) from adult mouse vestibular tissue revealed increased chromatin accessibility in the promoter regions of genes associated with

motile cilia machinery (*Figure 5—figure supplement 4*), suggesting elevated transcriptional activity in vestibular HCs. These findings are consistent with our observations from scRNA-seq datasets.

To examine the conservation of expression of these genes in vestibular HCs across species, we obtained the orthologs of axoneme-related genes in adult zebrafish inner ear HCs and human vestibular HCs using published datasets (*Barta et al., 2018*; *Wang et al., 2024*). *Figure 6A* shows the expression of these genes in adult mouse, zebrafish, and human vestibular HCs. While the expression levels vary, most of these genes are expressed across species. The exceptions are *Dnah3* and *Dnah12*, which are expressed in zebrafish HCs but not in mammalian vestibular HCs.

We conducted immunostaining and high-resolution confocal imaging to validate the expression of key motile cilia markers in the kinocilia. FOXJ1 is expressed in the nucleus of vestibular HCs, while CCDC39, CCDC40, TEKT1, DNAH5, and DNAH6 are expressed in kinocilia (*Figure 6B*). Collectively, our findings provide evidence corroborating the presence of motile cilium machinery in the kinocilia of vestibular HCs.

Since nascent cochlear HCs possess kinocilia (*Figure 6C*), we used published P2 cochlear and vestibular HC transcriptomes (*Burns et al., 2015*; *McInturff et al., 2018*) to investigate whether neonatal cochlear HCs express motile cilium-related genes. Assessment of expression of genes related to motile cilia machinery revealed a less drastic difference between neonatal cochlear and vestibular HCs than that between adult cochlear and vestibular HCs (*Figure 6D*). We note that proteins of some shared genes are expressed in both cilia and cytosol (marked by red asterisks in *Figure 6D*). However, some key motility-associated genes such as *Dnah6* and *Dnah5* (marked by red arrows in *Figure 6D*) were not detected in the P2 cochlear HCs. These axonemal dynein heavy chains are ATPase-driven force-generating motors that produce the ciliary power stroke in concert with other axonemal components. Immunostaining confirmed the lack of expression of CCDC39, CCDC40, and DNAH6 in cochlear HCs at P2 (*Figure 6E*). In contrast, these key proteins were expressed in kinocilia of vestibular HCs (*Figure 6E*). Lack of expression of *Dnah5* and *Dnah6* and the molecular rulers CCDC39 and CCDC40 suggests that the kinocilium of neonatal cochlear HCs does not possess signatures of motile cilia. Therefore, our analysis indicates that the molecular composition of kinocilia is different between neonatal cochlear and vestibular HCs.

## Vestibular kinocilia exhibit hybrid morphological features of primary and motile cilia

TEM studies have characterized the ultrastructure of kinocilia across species, revealing evidence of complex and regionally specialized organization (*Kikuchi et al., 1989*; *Nagel et al., 2014*; *O'Donnell and Zheng, 2022*). We used TEM to examine the ultrastructure of kinocilia from bullfrog crista HCs. TEM images, including longitudinal sections of frog vestibular hair bundles (*Figure 7A*), highlight a distinct zonal architecture along the kinocilium axis. At the distal tip, a prominent kinociliary bulb is observed, while the base anchors the axoneme within the cuticular plate. Two pairs of microtubule doublets span the full length of the kinocilium in the longitudinal section. The central pair of singlet microtubules, typical of motile cilia, is maintained throughout most of the shaft but disappears in the distal and transitional zones. This configuration results in a dynamic shift from a canonical 9 + 2 arrangement centrally to a 9 + 0 pattern at both the base and tip. These observations underscore the heterogeneous and hybrid nature of vestibular kinocilia, integrating structural features of both primary and motile cilia.

## Evidence that vestibular kinocilia exhibit motility

Vestibular kinocilia are traditionally regarded as non-motile, lacking the rhythmic beating characteristic of respiratory cilia. Kinocilia are not only tightly connected to stereocilia but are also embedded in the overlaying gelatinous membrane (*Eatock and Songer, 2011*; *Leibovici et al., 2005*; *Li et al., 2008*), an extracellular matrix required for physiological mechanotransduction. Using acute preparations of bullfrog semicircular canal sensory epithelia (cristae), we observed robust spontaneous kinociliary motility in some HCs (*Figure 7B*; *Videos 1 and 2*). This motility exhibited the characteristic beating pattern of flagella and cilia and occurred at a frequency of approximately 5–10 Hz at room temperature. The observed displacements were sufficiently large to induce deflection of the entire hair bundle, indicating it can generate forces to influence hair bundle dynamics. Interestingly, this phenomenon was detected in only ~1–5% of crista HCs, likely due to variable preservation of HC

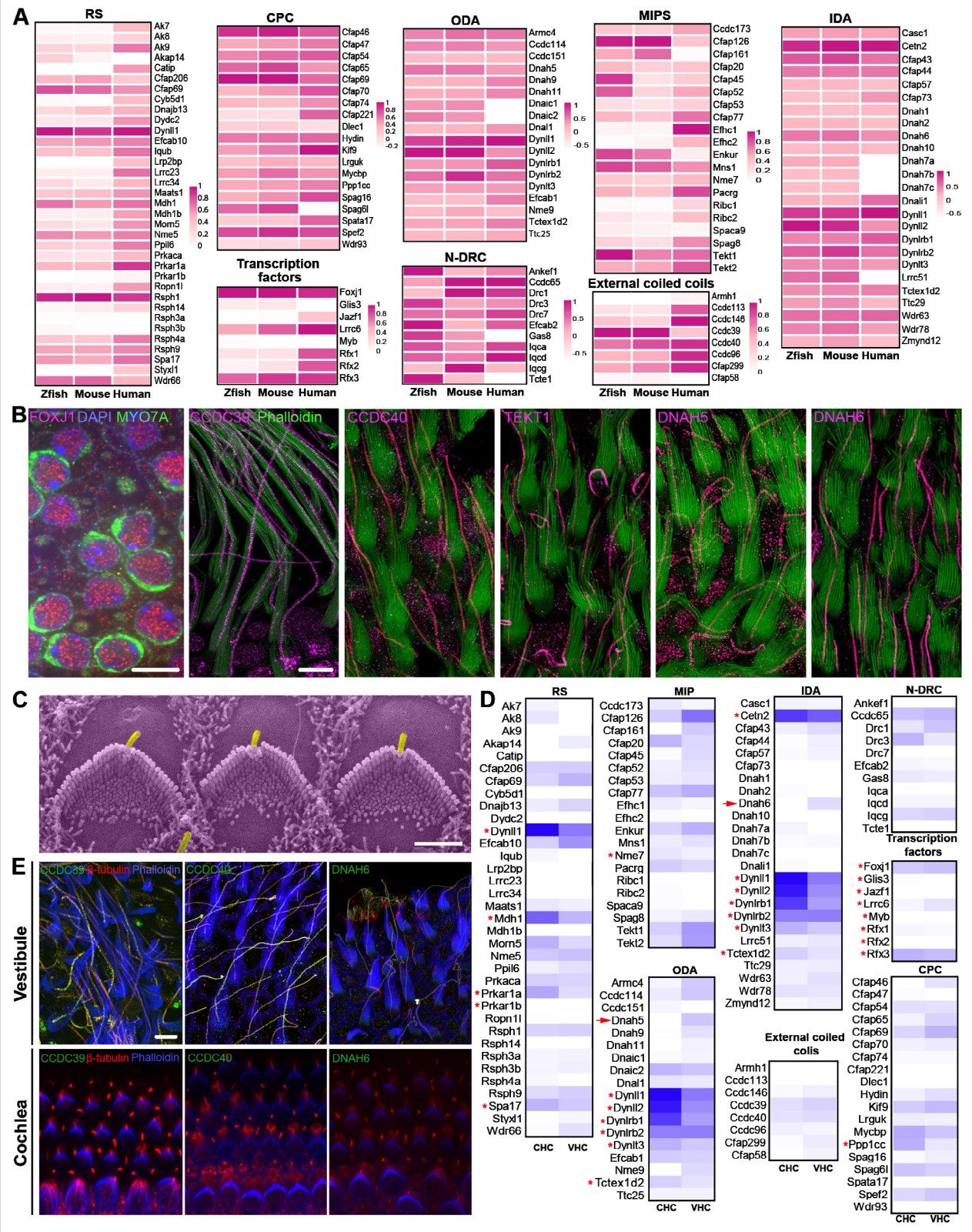

**Figure 6.** Expression of motile cilia-related genes/proteins in the vestibular HCs. (**A**) Expression of motile cilia-related genes in zebrafish, mouse, and human vestibular HCs. Expression values of these genes are based on *Supplementary file 1* (mouse), Data Citation 4 (*Barta et al., 2018*; GSE101693, zebrafish), single-cell RNA-sequencing (scRNA-seq) dataset (*Wang et al., 2024*, GSE207817, human). Mouse gene nomenclature is used in heatmaps. (**B**) Confocal images of the expression of key motile cilia-related proteins. Scale bars represent 5 µm. (**C**) SEM micrograph of hair bundles of OHCs

*Figure 6 continued on next page*

*Figure 6 continued*

from P2 cochlea. Kinocilia (in magenta) are still present at this age. Bar: 2.5 µm. (**D**) Comparison of expression of motile cilia-related genes between P2 cochlear and vestibular HCs. Gene expression values are based on HC transcriptomic dataset by *Burns et al., 2015*. Red asterisks mark the genes whose encoded proteins are expressed in both cilia and cytoplasm or multifunctional. Red arrows indicate *Dnah5* and *Dnah6*, which were not detected in P2 cochlear HCs. (**E**) Confocal images of expression of CCDC39, CCDC40, and DNAH6 in P2 vestibular and cochlear HCs. CCDC39, CCDC40, and DNAH6 were not expressed in cochlear HCs at P2. Bar: 5 µm.

and kinociliary integrity in vitro in acutely dissected tissue. These findings suggest that vestibular kinocilia are capable of active movement and challenge the strict classification of these structures as non-motile even though motility is not experimentally observed in most HCs in the excised vestibular sensory epithelium.

Next, we explored whether the kinocilia of vestibular HCs from adult mice are motile by utilizing the photodiode technique to detect bundle motion (*Jia and He, 2005*). This technique can detect motion in the 10 nm range for synchronized signals after averaging. Since spontaneous motions are not synchronized and cannot be averaged to improve the signal-to-noise ratio, we captured the responses in the time domain and averaged them in the frequency domain after a fast Fourier transform. This allows us to detect unsynchronized cilia motion even if the signal is close to the noise level at the time domain. We first measured spontaneous movement of cilia from the epithelial lining of the Eustachian tube, which is an extension of airway epithelia, as cilia in the respiratory tract are a typical example of motile cilia (*Li et al., 2014*). *Figure 7C* shows two representative waveforms of airway cilia beat with a magnitude between 700 and 1500 nm. The two responses shown in *Figure 7C* have main frequency components at 6–9 Hz at room temperature, consistent with a previous study (*Li et al., 2014*). Next, we measured the movement of the top segment of the hair bundle from mouse crista ampullaris. Since the kinocilium is tightly attached to the stereocilia bundle, we measured the motion of the whole bundle due to the difficulty of taking measurements from a single kinocilium. The waveform (black trace in *Figure 7D*) was obtained from crista HCs bathed in perilymph-like solution (L-15 medium) with 2 mM of $Ca^{2+}$. Like the beat frequency of airway cilia, kinocilia also moved at the frequency of ~7 Hz (*Figure 7D*). To rule out the possibility that the bundle motion is driven by the mechanotransduction-related activity (*Benser et al., 1996*; *Martin et al., 2003*), we treated the crista ampullaris in $Ca^{2+}$-free medium with EGTA for 2 min to break the tip-link (*Assad et al., 1991*; *Jia et al., 2007*; *Kachar et al., 2000*; *Ricci et al., 2003*). Spontaneous motion was still detected (two red traces in *Figure 7D*), suggesting that the motion is independent of the transduction channel activity. We measured spontaneous bundle motions from 52 crista HCs from six mice. Spontaneous motion was only detected in eight HCs. An example of a lack of response is shown in *Figure 7D* (blue trace). Although we were unable to determine which of the HC subtypes were exhibiting kinocilia motility, it is conceivable that both type I and II HCs have this capability since they both express the genes related to motile cilia. We note the kinocilia motion of mouse crista HCs was substantially smaller than that of airway cilia (*Figure 7D*) and bullfrog crista HCs.

## Predicted model of the 96-nm modular repeat in adult vestibular kinocilia

Since kinocilia motility in mouse vestibular HCs is substantially smaller than the motility of airway cilia, we investigated the structural basis underlying comparatively reduced motility of kinocilia. This diminished motility may result from differences in the molecular composition and organization of the axonemal machinery, particularly the 96 nm modular repeat that houses key dynein motors and regulatory complexes. Recent advances in structure prediction powered by artificial intelligence and cryo-EM have facilitated the generation of highly conserved atomic models of the 96 nm axonemal repeat from human respiratory cilia and bovine sperm flagella (*Chen et al., 2023*; *Leung et al., 2025*; *Walton et al., 2023*). We applied our axonemal gene dataset to these atomic models to predict the molecular architecture of the 96 nm repeat in vestibular kinocilia. By mapping the expression of known axonemal components onto the structural frameworks derived from human respiratory (PDB: 8J07) (*Walton et al., 2023*) and bovine sperm (PDB: 9FQR) (*Leung et al., 2025*) axonemes, we generated two composite models that reflect the unique molecular composition of the vestibular kinocilium. These predicted structures are shown in *Figure 8A, B*. 3D videos of the predicted models are provided in *Videos 3 and 4*.

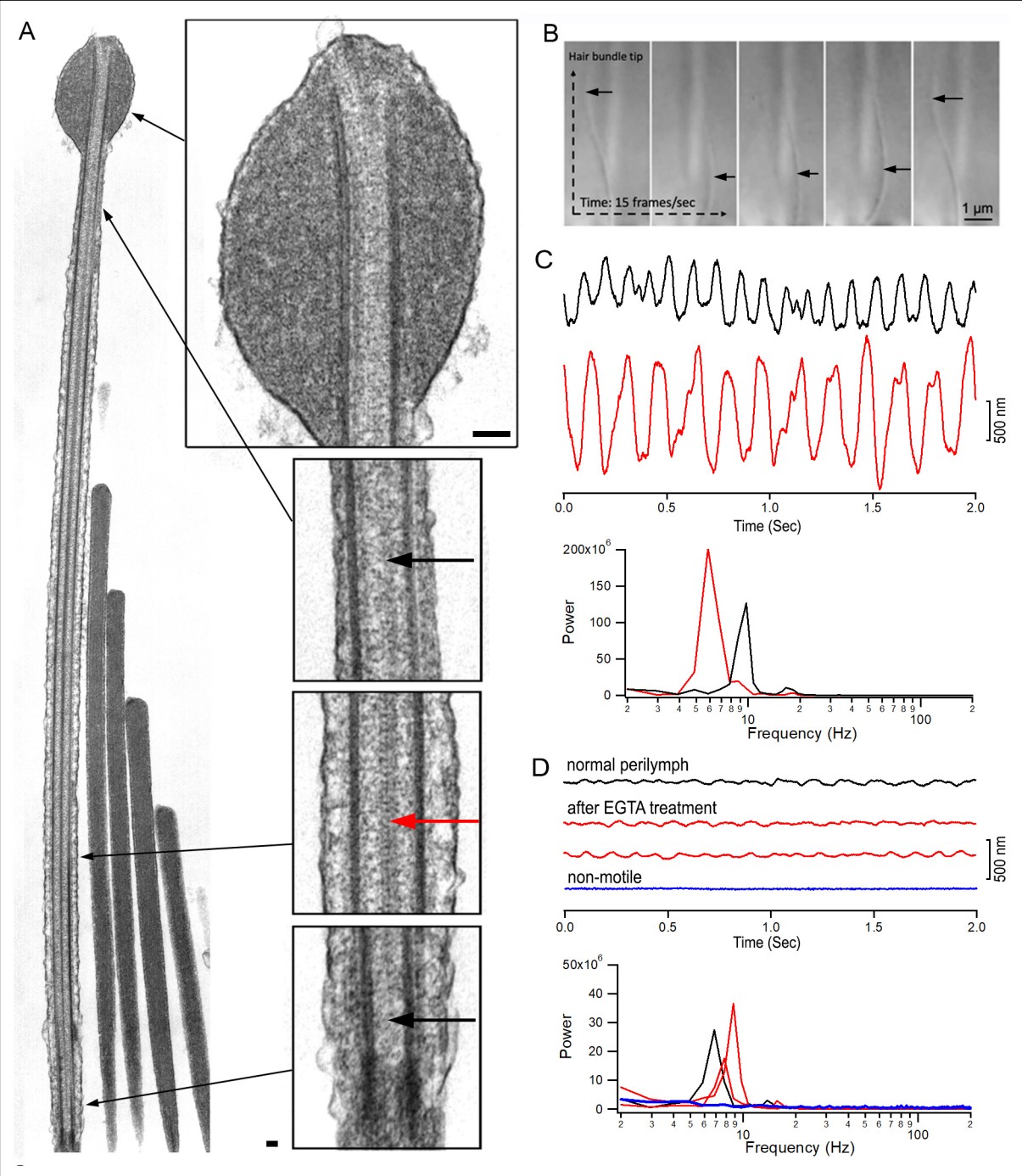

**Figure 7.** Kinocilia morphology and motility. (**A**) Transmission electron microscopy (TEM) images of stereocilia and kinocilium from bullfrog crista HCs. Different regions of the kinocilium in higher magnification are also shown. Long black arrows indicate where the magnified images were taken. Bars: 250 nm. Red arrow indicates two central microtubule singlets. Short black arrows mark the absence of central microtubule singlets in the distal regions near the tip of kinocilium and transition zone. (**B**) Images captured from in vitro live imaging of kinocilium and bundle motion of a bullfrog crista HC. The images were captured at a speed of 15 frames per second. Black arrows indicate kinocilium. (**C**) Representative waveforms of spontaneous cilia motion from middle ear tissue. The FFT analysis of cilia motion is also shown. (**D**) Three representative waveforms of spontaneous motion of hair bundles. The response waveform in blue was taken from a hair bundle with no spontaneous motility. FFT analysis of bundle motion is shown. Response waveforms and spectra are color-coded and -matched.

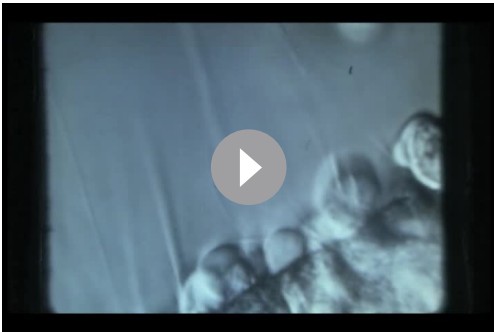

**Video 1.** Bullfrog kinocilia motility.
https://elifesciences.org/articles/108071/figures#video1

We chose the human respiratory 96 nm axonemal structure as our reference because it best reflects vestibular kinocilia and provides a more comprehensive representation of axonemal components than the sperm model, which shows more missing elements (highlighted in gold in *Figure 8A, B*). Using a curated reference gene list derived from the human respiratory 96 nm axonemal structure, we mapped vestibular HC gene expression and identified transcripts for all 18 ODA genes and their docking complex components, all 11 N-DRC genes, and all 7 MAPs, along with 36 of 37 RS genes and 19 of 23 IDA-related genes. Additionally, 20 of the 31 genes encoding MIPs, which are known to stabilize the microtubule doublets, were also expressed (*Figure 5G*). In contrast, vestibular HCs lacked the expression of several sperm-specific genes from distinct compartments of 96 nm repeat (*Leung et al., 2025*), including TRiC chaperonin subunits (*Brown et al., 2025*; *Meng et al., 2026*), *Camk4*, *Efacb5*, *Lrrd1*, *Stkld1*, *Ccdc63*, *Wdr64*, and several MIPs (*Supplementary file 3*).

Based on our models, the absence of *Dnah3* and *Dnah12*, along with *Acta2*, is predicted to result in the loss of two of the six single-headed IDA components (highlighted in gold). *Cfap100*, which forms the modifier of inner arms (MIA) complex with *Cfap73* and contributes to tethering the double-headed IDAf (inner dynein arm f), is missing in the model, whereas *Cfap73* and the remaining IDAf components are present. Notably, IDAf has multiple attachment points, and MIA is not essential for docking IDAf to the DMTs (*Yamamoto et al., 2013*). Overall, 4 of the 23 IDA-related genes were not expressed in vestibular HCs; nonetheless, we predict that the IDA structure is reduced but not entirely absent, as DNAH6 remains localized in vestibular kinocilia (*Figure 5B*). In addition to the missing IDA components, we identified 11 unexpressed genes associated with MIPs, whose absence is predicted to result in reduced MIP density in the models (highlighted in orchid and gold in cross-sectional views in *Figure 8A, B*). Unlike other axonemal structures, MIPs exhibit greater variability across species, which may account for their lineage-specific absence in vestibular kinocilia (*Figure 8*; *Andersen et al., 2024*; *Tai et al., 2023*; *Xia et al., 2025*). The missing genes in our HC datasets and their roles in the axonemal complex, cilia motility, and ciliopathy are listed in *Figure 8C* and *Supplementary file 3*. Based on our predicted models, we speculate that the absence of *Dnah3* and *Dnah12* plays a major role in limiting kinocilia motility in mouse vestibular HCs, contributing to the smaller movements compared to respiratory motile cilia.

## Discussion

This is the first study to compare transcriptomes among four types of HCs from the adult mouse inner ear and to characterize the molecular composition of kinocilia. We found that the transcriptomic similarity between type I and II HCs is greater than that between IHCs and OHCs, indicating greater homogeneity among vestibular HC subtypes compared to cochlear HC subtypes. We identified several new genes and proteins that can be used as markers for vestibular HCs, especially those related to kinocilia. We observed notable differences in gene signatures related to HC unique structure and function, which may underlie distinct biological properties of mechanotransduction, membrane conductance, and synaptic transmission seen among the four different HC types. Differential expressions also explain why loss of function of a gene such as *Tmc1*, *Cib2*, or *Cib3* leads to differential auditory and vestibular phenotypes in mouse models and humans. Our

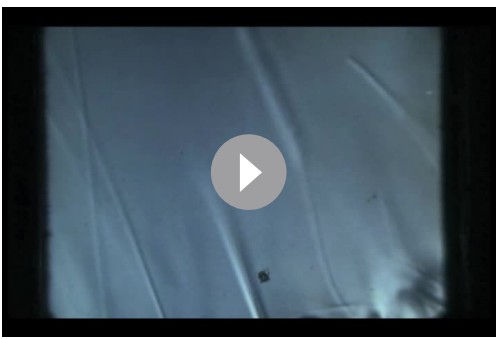

**Video 2.** Bullfrog kinocilia motility.
https://elifesciences.org/articles/108071/figures#video2

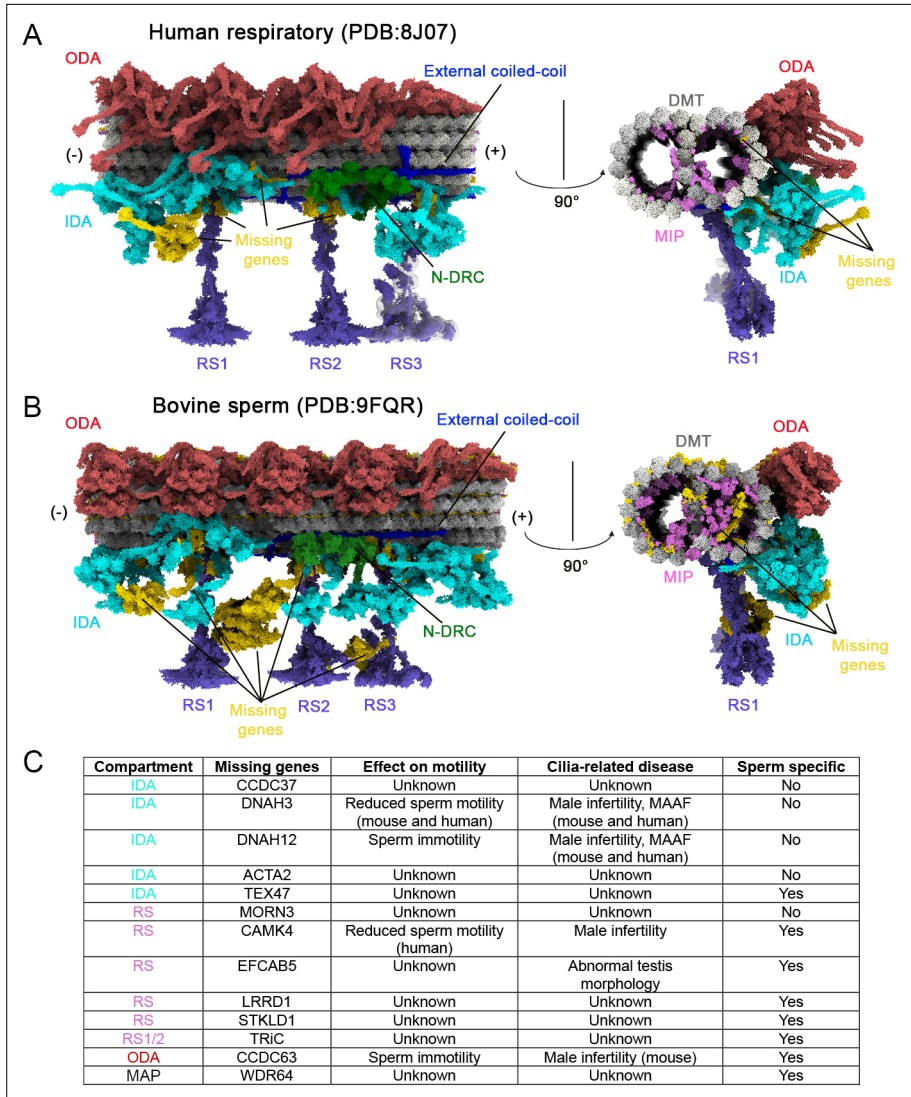

**Figure 8.** Predicted models of the molecular architecture of 96 nm axonemal repeat of vestibular kinocilia. Longitudinal and cross-sectional views of the doublet microtubule (DMT) and associated structure in 96 nm repeat, derived from combining cryo-electron microscopy (cryo-EM) data and single-cell transcriptomic analysis from human respiratory cilia (**A**) and bovine sperm flagella (**B**). Key axonemal motile-machinery components are color-coded: ODA (Indian red), IDA (cyan), N-DRC (green), MIPs (orchid), RS (purple), and external coiled-coils (blue). Radial spoke 3 (RS3) has not been resolved to atomic resolution, but its shorter form (RS3s) is depicted. DMTs are represented in gray. Regions highlighted in gold indicate the absence of corresponding transcripts in our mouse transcriptomic data. (**C**) Genes which are not detected in mouse and human vestibular HC transcriptomes and related to motility-relevant compartments are listed in the table. The roles of these genes in the 96 nm repeat module and cilia motility and ciliopathy are also included.

dataset is expected to serve not only as a valuable resource for unraveling the molecular mechanisms of the biological properties of HCs but also for assisting the auditory and vestibular research community in identifying and exploring the functions of disease-related genes.

Since biological processes enriched in vestibular HCs are related to cilia and cilia motility, we focused our analyses on kinocilia. Although kinocilium has long been considered a primary cilium, its molecular composition and structural organization remain largely unexplored. Our study suggests that kinocilia serve dual roles as both primary and motile cilia. The primary cilium is a major hub in receiving and transmitting signals from the environment. A recent study has demonstrated the expression of proteins related to transduction pathways and receptors in primary cilia using spatial proteomics. In line with this, we found that both vestibular and cochlear HCs express genes encoding

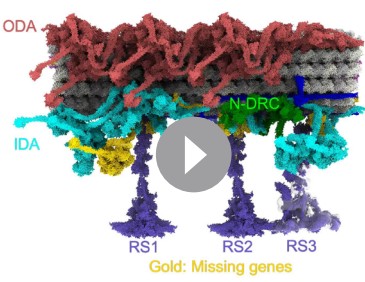

**Video 3.** Predicted structures models of molecular composition of the vestibular kinocilium based on the structural frameworks derived from human (Video 3) respiratory (PDB: 8J07) (*Walton et al., 2023*) and bovine (*Video 4*) sperm (PDB: 9FQR) (*Leung et al., 2025*) axonemes.

https://elifesciences.org/articles/108071/figures#video3

components of signal transduction pathways and receptors. While the precise localization of these proteins within HC kinocilia remains to be validated, our analysis reveals a shared expression of diverse primary cilium signaling genes across HC subtypes. This suggests a conserved, and potentially specialized, role for kinocilia in cellular signaling and ciliary function.

Although adult cochlear HCs lack kinocilia, we still observed expression of numerous cilia-related genes in these cells (*Figures 5 and 6*). While some of these genes may be vestigial, many are associated with primary cilia structures, including the basal body and IFT machinery. Notably, the basal body persists in adult cochlear HCs despite the developmental disappearance of the kinocilium. Previous studies using cilia proteomics have shown that many cilia-related proteins are expressed in cytosol, including proteins related to signal transduction, microtubule cytoskeleton, actin cytoskeleton, vesicle transport, metabolism, protein folding, translation, nuclear transport, ubiquitination, RNA binding, mitochondria, and transcriptional regulation (*van Dam et al., 2019*; *Pir et al., 2024*). Therefore, it is not unexpected that adult cochlear HCs continue to express genes associated with ciliary functions.

A TEM study of mouse vestibular kinocilia (*O'Donnell and Zheng, 2022*) showed the '9 + 2' microtubule arrangement characteristic of motile cilia. This contrasts with an earlier study of guinea pig vestibular kinocilia, which reported the absence of central singlet microtubules and IDAs, while ODAs and RSs were present (*Kikuchi et al., 1989*). In our study, TEM revealed a complex and regionally specialized organization of the kinocilium in bullfrog vestibular HCs. The axoneme is anchored within the cuticular plate at the kinocilium base. While the central pair of singlet microtubules is preserved along most of the shaft, it is absent in the distal and transitional zones—indicating a transition from the canonical 9 + 2 microtubule arrangement to a 9 + 0 configuration at both the base and tip. In most motile cilia, the central pair does not originate directly from the basal body; instead, it begins a short distance above the transition zone, a feature that illustrates variation across systems (*Lechtreck et al., 2013*). The central pair can also show variation in its spatial extent: for example, in mammalian sperm axonemes, it can terminate before reaching the distal end of the axoneme (*Fawcett and Ito, 1965*). In addition, the central pair orientation differs across organisms: in metazoans and *Trypanosoma*, the central pair is fixed relative to the outer doublets, whereas in *Chlamydomonas* and ciliates it twists within the axoneme (*Lechtreck et al., 2013*). Such structural variation has been observed in various motile cilia and flagella and is therefore not unique to vestibular kinocilia. However, a more distinctive feature of kinocilium morphology is the organization at the distal tip, where a prominent distal head is present—resembling tip structures recently identified in human islet cell cilia (*Polino et al., 2023*). This distal-most region is known to harbor specialized proteins (*Legal et al., 2023*). In multi-ciliated cells, CCDC33 and CCDC78 are found at the very end of the cilium and help organize other proteins like SPEF1, CEP104, and EB3/MAPRE3 (*Hong et al., 2025*; *Legal et al., 2025*; *Legal et al., 2023*). We observed expression of all genes encoding these proteins, except for

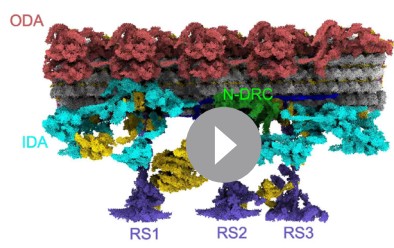

**Video 4.** Predicted structures models of molecular composition of the vestibular kinocilium based on the structural frameworks derived from bovine sperm.

https://elifesciences.org/articles/108071/figures#video4

*Ccdc78*. Although we did not study the tip of the kinocilium, its bulb shape suggests it may also contain specialized proteins. In bullfrog HCs, the kinocilial bulb binds to the overlaying otoconial membrane (*Jaeger et al., 1994*; *Kachar et al., 1990*), and shows strong labeling for β-tubulin and cadherin 23 (*Jaeger et al., 1994*; *Kachar et al., 1990*; *Lagziel et al., 2005*), and a recent study showed that *saxo2* overexpression in zebrafish results in bulbed kinocilia, with *saxo2* protein accumulation at the distal tip (*Erickson et al., 2023*). These bulbed tips may reflect specialized regulation of ciliary cap proteins (*Legal et al., 2023*) that help organize or stabilize the plus ends of axonemal microtubules—an area that remains to be explored in kinocilia.

The most important and novel finding of our study is that adult mouse vestibular HCs express genes related to the 96 nm axonemal repeat complex, a hallmark structural feature of motile cilia. Notably, orthologs of these genes are also expressed in zebrafish and human vestibular HCs, highlighting the evolutionary conservation of this molecular complex across vertebrate species. Furthermore, we observed robust spontaneous kinociliary motility in bullfrog crista HCs, as well as subtle spontaneous bundle movements in mouse crista HCs. Our findings indicate that this motion is independent of the mechanotransduction apparatus, as the kinocilium itself exhibited active, flagellar-like movement in bullfrog crista HCs (*Videos 1 and 2*). In mouse crista HCs, spontaneous motion was still present after breaking the tip links. Early studies reported observation of spontaneous flagella-like rhythmic beating of kinocilia in vestibular HCs in frogs and eels (*Flock et al., 1977*; *Rüsch and Thurm, 1990*), as well as in zebrafish HCs in the early otic vesicle (*Stooke-Vaughan et al., 2012*; *Wu et al., 2011*). According to Rüsch and Thurm, spontaneous kinociliary motility was observed only under conditions of tissue deterioration. They therefore interpreted kinocilia beating as a sign of cellular decline rather than a physiological feature. We speculate that deterioration may have disrupted kinocilial links, effectively unloading the kinocilium and permitting freer movement. Nonetheless, regardless of tissue condition, the observation of spontaneous kinocilia beating along with expressions of motile cilia signature genes and proteins supports the conclusion that kinocilia are motile cilia.

We observed kinociliary beating in only a subset of the cells. While we cannot exclude the possibility that indeed only some kinocilia are inherently motile, or that stressful conditions may activate a latent motility in vestibular HCs, there are several possible reasons why such motility has not been consistently observed in most vestibular HCs—both in our study and in previous investigations. For example, a reduction in intracellular ATP levels in in vitro preparations may play a significant role, as ciliary motility is ATP-dependent. Rapid depolarization of HCs in vitro is an indication of reduced availability of intracellular ATP, as $Na^+/K^+$-ATPase pump is critical for maintaining resting membrane potential (*He and Dallos, 1999*; *Silver and Erecińska, 1997*). Additionally, acute tissue dissection may disrupt kinocilia, which are normally straight and tethered to the extracellular matrix. In vitro, many appeared bent, potentially compromising their structural integrity and the metabolic conditions required for motility.

Although we observed kinocilium-driven bundle movement in adult mouse vestibular HCs, its magnitude was at least an order of magnitude lower than that of other motile cilia. Our single-cell transcriptomic analysis showed the absence of 16 genes associated with the 96 nm axonemal repeat of human respiratory cilia, including *Pierce1* and *Pierce2*. These two MIP-encoding genes have been shown to regulate motile cilia function and left–right asymmetry in mouse models (*Gui et al., 2021*; *Sung et al., 2016*). Knockout of *Pierce1* results in pronounced defects in ciliary motility and dynein arm docking, while *Pierce2* loss has a milder effect and largely preserves overall ciliary ultrastructure and beating (*Gui et al., 2021*; *Sung et al., 2016*). Although the contribution of these genes to kinociliary function remains uncertain, the absence of both may contribute to reduced microtubule stability or motor organization, especially when combined with additional losses in key motor components. However, MIPs are the most heterogeneous components across different types of cilia, such as sperm and airway cilia in different species (*Leung et al., 2025*; *Tai et al., 2023*). Thus, it remains unclear whether the absence of these two genes reduces kinociliary motility in mouse vestibular HCs.

The lack of expression of two genes associated with IDAs (*Dnah12* and *Dnah3*) may lead to the loss of specific single-headed IDA components, as suggested by our structural models (*Figure 8*). Mutations of *DNAH3* and *DNAH12* are linked to male infertility and dynein dysfunction in humans (*Meng et al., 2024*; *Yang et al., 2024*). Mutations of DNAH12 cause male infertility by impairing DNAH1 and DNALI1 recruitment. However, it does not affect the tracheal tract and oviductal cilia organization (*Yang et al., 2024*). For the other two IDA-related genes (*Acta2*, *Cfap100/Ccdc37*) and

one RS-related gene (*Morn3*), no ciliopathy has been linked to mutations of these three genes so far. We speculate that the lack of expression of DNAH3 and DNAH12 may be a key factor limiting the magnitude of kinocilia motility in mammalian vestibular HCs compared to respiratory cilia or kinocilia of bullfrog vestibular HCs.

We detected the expression of a few sperm-specific MIPs including *Saxo4*, *Tekt3*, and *Tekt4* in vestibular HCs. While the kinocilium shares a broader molecular profile with epithelial motile cilia, the presence of these distinct sperm-specific MIPs, which are absent from multi-ciliated epithelial tissues, suggests kinocilia may possess unique structural specializations adapted to their exceptional length (~60–70 μm in length in mouse crista HCs) and sensory function, highlighting the unique identity of kinocilia.

HCs employ positive local feedback to amplify inputs to their mechanosensitive hair bundles (*Fettiplace, 2017*; *Hudspeth, 1997*). This amplification helps overcome mechanical impedances and fine-tune sensory stimuli. In mammals, the remarkable sensitivity of the auditory system is largely attributed to the fast somatic motility of OHCs in the cochlea (*Brownell et al., 1985*; *Dallos et al., 2008*; *Kachar et al., 1986*; *Liberman et al., 2002*; *Zheng et al., 2000*). In other receptor organs, HCs may effect amplification by the $Ca^{2+}$-dependent activity of myosin or transduction channels in the stereocilia (*Fettiplace, 2017*; *Hudspeth, 1997*). Mechanotransduction-mediated active hair bundle movements have been reported in turtle and frog HCs (*Benser et al., 1996*; *Crawford and Fettiplace, 1985*; *Denk and Webb, 1992*; *Howard and Hudspeth, 1987*; *Martin et al., 2003*). The functional significance of kinociliary beating remains to be elucidated; however, the kinocilium may serve as an active, force-generating component of the hair bundle. Because the kinocilium is connected to the tallest stereocilia via kinocilial links, we speculate that kinociliary motility may dynamically modulate the mechanical properties of the hair bundle or influence tip-link tension to prime transduction channels. Kinociliary beating is sufficient to drive stereocilia bundle movement (*Videos 1 and 2*). Even when constrained by the overlying otolithic membrane or cupula, changes in kinociliary stiffness could still affect the bundle's mechanical dynamics. Importantly, such autonomous rhythms are unlikely to disrupt temporally accurate encoding of head motion, as spontaneous bundle movements driven by mechanotransduction have also been observed in bullfrog saccular HCs (*Benser et al., 1996*; *Martin et al., 2003*). Moreover, auditory and vestibular afferent neurons also generate spontaneous action potentials in both developing and mature animals. Although we did not examine when spontaneous kinocilia beat emerges during development, our analysis showed that key motile cilia signature genes and proteins are expressed at P2, suggesting that spontaneous kinocilia beat may already be present at this age. Such activity may help refine HC maturation and neural connections and prime the vestibular central pathway for its later function.

In summary, this study demonstrates that the kinocilium of vestibular HCs is a unique hybrid cilium, exhibiting strong overlap of molecular features of both primary and motile cilia. While it shares structural and molecular similarities with motile cilia and sperm flagella, it also possesses distinct architectural and functional characteristics. Future investigations employing kinocilium-specific proteomics, cryo-ET, and single-particle analysis will be critical for fully characterizing kinocilium molecular composition and organization. Although kinocilia motility was observed in bullfrog and mouse vestibular HCs in the present study, the functional significance of kinocilia motility remains to be elucidated.

## Materials and methods

Male and female CBA/J mice were purchased from the Jackson Laboratory (Stock #:000656) and reared in the Animal Care Facility of Creighton University and NIDCD. American bullfrogs (*Rana catesbiana*) were purchased from Carolina Biological Supply Co. The animal usage and care were approved by the Institutional Animal Care and Use Committees of Creighton University (Protocol #23001046) and NIDCD (NIDCD ACUC Protocol #1215).

### Cell dissociation, cDNA libraries preparation, and RNA-sequencing

Male and female CBA/J mice aged 10 weeks were used for scRNA-seq. Cochlear and vestibular end organs (utricle, saccule, and crista) were dissected from the inner ear and placed in Petri dishes containing cold L-15 medium (Gibco; #11320033). After the cochlear and vestibular sensory epithelia and neurons were dissected out, they were transferred into two individual 1.5 ml tubes for enzymatic

digestion (Collagenase IV from Sigma, concentration: 1 mg/ml collagenase) in L-15 medium. After 10 min of incubation at room temperature, the enzymatic solution was removed and replaced with 400 μl L-15 media containing 10% fetal bovine serum. The tissues in two tubes were mechanically triturated by 200 μl Eppendorf pipette tips. After that, the suspension containing cochlear and vestibular cells was then passed through 40 μm strainers for filtration and pelleted at 300 × *g* for 5 min. After removing extra media, cells were then reconstituted in the 50 μl L-15 with 10% fetal bovine serum media and used for cDNA library preparation. Seven mice were used for each biological replicate. Six biological replicates for cochlear sensory epithelium and four biological replicates for vestibular sensory epithelia were prepared for scRNA-seq.

The emulsion droplets were constructed using a 10x Genomics Controller device following the manufacturer's instruction manual. cDNA libraries were constructed using the 10x Genomics Chromium Single Cell 3′ Reagent Kits V3.1. High Sensitivity DNA Kits (Agilent Technologies) were used to perform quality control for each library in an Agilent 2100 Bioanalyzer. cDNA libraries were sequenced in an Illumina NextSeq 6000 sequencer aiming for 240 billion 150 bp long paired-end reads.

### Single-cell RNA-seq data processing and analysis

Raw transcriptomic datasets of adult cochlear and vestibular HCs from scRNA-seq have been deposited to GEO (GSE283534). The FASTQ files were mapped to mm10 reference genome to generate the single-cell expression matrices following the CellRanger count pipeline (version 6.1.2). The Cellranger output data was then processed with the Seurat package (version 4.3.0) in R (version 4.1.3).

Genes expressed in at least ten cells were included in the analysis. Cells with numbers of expressed genes <200 or >3000 and cells with numbers of unique molecular identifiers >15,000 were filtered out. Cells with >20% mitochondrial genes were also excluded from the analysis.

The gene expression data from individual samples were converted into a natural logarithm and normalized under the same condition. Data from six cochlear and four vestibular replicates were integrated separately based on the anchors identified from the top 2000 highly variable genes of individual normalized expression matrices. The Shared Nearest Neighbor graph method can calculate the neighborhood overlap (Jaccard index) between every cell and its nearest neighbors, which was used for cluster determination at a resolution of 0.6 on PCA-reduced expression data for the top 30 principal components.

Clustering results for cochlear and vestibular datasets were visualized separately using t-SNE. Cluster annotations were initially produced using SingleR and then corrected where appropriate based on well-known cellular markers for cochlear and vestibular cells as described before (*Xu et al., 2022*).

Entrez Gene, HGNC, OMIM, and Ensembl database were used for verification, reference, and analyses. Online Databases of Ciliogenics (https://ciliogenics.com/), CiliaCarta (https://ngdc.cncb.ac.cn/databasecommons/database/id/6383), and Primary Cilium Proteome (https://esbl.nhlbi.nih.gov/Databases/CiliumProteome/) were also used for reference.

GO analysis in *Figure 5—figure supplement 1* was performed using ShinyGO 0.82 (*Ge et al., 2020*). The Venn diagrams in *Figure 5B*, *Figure 5—figure supplement 3* were generated using Venny 2.1 (https://bioinfogp.cnb.csic.es/tools/venny/), except for the sperm Venn diagram in *Figure 5—figure supplement 3E*, which was plotted using nVenn (https://degradome.uniovi.es/cgi-bin/nVenn/nvenn.cgi). *Figure 5C* was created with BioRender.com and further modified using Adobe Photoshop.

### Immunocytochemistry

Inner ears were fixed overnight with 4% paraformaldehyde at 4°C. Cochlear and vestibular sensory epithelia were dissected out. After several washes with PBS, the tissue was blocked for 1 hr with 0.25% normal goat serum in PBS containing Triton X-100 (0.01%) and goat serum (10%). Primary antibodies against DNM1 (NBP2-48950, Novus Biologicals), SLC7A14 (HPA045929, Sigma), TJAP1 (NBP1-80902, Novus), FOXJ1 (14-9965-82, Thermo Fisher), CCDC39 (HPA035364, Sigma), CCDC40 (PA5-54653, Thermo Fisher), DNAH5 (31079-1-AP, Thermo Fisher), DNAH6 (HPA036391, Sigma), TEKT1 (HPA044444, Millipore Sigma), CLUAP1 (PA5-83710, Thermo Fisher), IFT172 (28441-1-AP), and acetylated tubulin (T6793, Sigma) were incubated with the tissues for 12 hr at 4°C. After washes with PBS, secondary antibody (1:500) (Alexa Fluor Molecular Probe 488 or 555; Invitrogen) was added and incubated for 2 hr at room temperature. Alexa Fluor 405 or 488 phalloidin (A30104 or A12379, Invitrogen) was used to label stereocilia bundles. Tissues were washed with PBS and mounted on glass

microscopy slides with antifade solution (5 ml PBS, 5 ml glycerol, 0.1 g *n*-Propyl gallate). Images were captured using a Nikon TI-2 Spinning Disk or Zeiss LSM 980 Inverted confocal microscope. Immunostaining of each type of antibody was repeated in four mice.

## Single-molecule fluorescence in situ hybridization

Single-molecule fluorescence in situ hybridization was used to validate the expression of 15 genes in 10-μm-thin sections prepared from three mice. Samples were prepared in formalin-fixed paraffin-embedded tissue. Probes for 18 genes were purchased from ACD. These genes include *Adam11* (Cat#: 580971), *Aqp11* (Cat#: 803751), *C1ql1* (Cat#: 465081), *Cdh23* (Cat#: 567261-C2), *Chrna10* (Cat#: 818521), *Cib2* (Cat#: 846681), *Cib3* (Cat#: 1105771), *Dnm3* (Cat#: 451841), *Ikzf2* (Cat#: 500001), *Kcnq4* (Cat#: 707481), *Otof* (Cat#: 485678), *Pcdh20* (Cat#: 467491), *Slc7a14* (Cat#: 544781), *Tmc1* (Cat#: 520911-C2), and *Tbx2* (Cat#: 448991-C2). Methods for the RNAscope 2.5 HD Duplex Assay from Advanced Cell Diagnostics were followed.

## Electron microscopy

The mouse inner ears were fixed with 4% paraformaldehyde and 2.5% glutaraldehyde in 0.1 M sodium cacodylate buffer (pH 7.4) with 2 mM $CaCl_2$. The bullfrog crista ampullaris was fixed with 3% para-formaldehyde, 2% glutaraldehyde, 2% tannic acid, and 0.5% calcium chloride in 0.1 M sodium cacodylate buffer (pH 6.8) with 0.1 mM $CaCl_2$ (*Kachar et al., 1990*). The tissues were then post-fixed for 1 hr with 1% $OsO_4$ in 0.1 M sodium cacodylate buffer and washed. For SEM, cochleae and vestibular tissues were dehydrated via an ethanol series, critical point dried from $CO_2$, and sputter-coated with platinum. The morphology of the HC stereocilia bundle was examined in a FEI Quanta 200 scanning electron microscope and photographed. For TEM, the bullfrog crista ampullaris was embedded in plastic (Epon 812 Epoxy Resin) after dehydration via an ethanol series. 70 nm thin sections were cut with a diamond knife and collected on 300-mesh grids. The thin section was post-stained with 3% uranyl acetate for 15 min and 1.5% lead citrate for 3 min. The preparations were examined in an electron microscope (JEOL 100CX) and photographed. Three animals were used for EM studies.

## Measurements of ciliary motion

Bullfrogs were anesthetized with 20 μg/g of 3-aminobenzoic acid ethyl ester and decapitated. The saccular macula was carefully dissected out under cooled frog's Ringer solution. The otoliths were gently removed with forceps in fresh Ringer solution (*Jaeger et al., 1994*). The saccular macula was placed under an Olympus upright microscope and kinocilia motility was visualized using DIC and a 60x objective and captured with a camera.

The Eustachian tube was dissected out from CBA/J mice and sectioned along its longitudinal length. The preparation was bathed in L-15 medium (Invitrogen), containing 136 mM NaCl, 5.8 mM $NaH_2PO_4$, 5.4 mM KCl, 1.4 mM $CaCl_2$, 0.9 mM $MgCl_2$, 0.4 mM $MgSO_4$, and 10 mM HEPES-NaOH (pH 7.4, 300 mmol/l) in an experimental chamber mounted on the stage of a Leica upright microscope. Crista ampullaris was also dissected from 10-week-old CBA/J mice and bathed in L-15 medium. The tissue was attached to the bottom of the chamber by the weight of two thin platinum rods (0.5 mm in diameter). The tissue was mounted with the cilia or hair bundles facing upward toward the water-immersion objective. The cilia and hair bundles were imaged using a 63x water immersion objective (Leica) and magnified by an additional 10x relay lens. Ciliary motion was measured and calibrated by a photodiode-based measurement system mounted on the Leica upright microscope (*Jia and He, 2005*). The magnified image of the hair bundle was projected onto a photodiode through a rectangular slit. The image was positioned to one side of the slit with 50% of the rectangular slit being covered by the magnified image of the bundle. Cilia motion modulated the light influx to the photodiode. The photocurrent response was calibrated to displacement units by moving the slit a fixed distance (0.5 μm) with the image of the cell in front of the photodiode. After amplification, the photocurrent signal was low-pass filtered by an antialiasing filter before being digitized by a 16-bit A/D board (Digidata 1550A; Molecular Devices). The motile responses were low-pass filtered at 250 Hz and digitized at 1 kHz. Ciliary motion was acquired in a 2-s window for each trial, and 20 trials were captured for each cell in one recording. The power spectrum of the response was averaged and analyzed in the frequency domain using Clampfit software (version 10, Molecular Devices). The experiments were performed at room temperature (22 ± 2°C).

## Predicted model of the 96 nm modular repeat in adult vestibular kinocilia

The 96 nm repeat structures of the human respiratory axoneme (PDB: 8J07) and bovine sperm flagellum (PDB: 9FQR) were used as structural references to model the vestibular kinocilia axoneme. Candidate vestibular HC genes identified through transcriptomic profiling were annotated and mapped to their respective axonemal compartments based on known or predicted protein localizations within these reference structures. The 8J07 model lacked atomic modeling for the full-length RS3 (*Zhao et al., 2025*), despite its presence in the corresponding cryo-EM density map (EMD-35888), due to the uncharacterized proteome of RS3 in the human respiratory system. To address this, RS3 components from the sperm axoneme structure (PDB: 9FQR), excluding sperm-specific proteins, were extracted, fitted into the EMD-35888 density using ChimeraX's 'fit-to-map' tool, and overlaid onto the 8J07 model to complete the RS3 architecture. All structural visualization and model integration were performed using UCSF ChimeraX v1.6. (*Pettersen et al., 2021*). In the models, DNAH5 and DNAH9 (present in our HC data) occupy the ODA region in human respiratory cilia, while DNAH8 and DNAH17 occupy the same region in sperm (but are not expressed in our dataset). Since they correspond structurally, we did not mark them as missing in the sperm-based kinocilia model.

## Code availability

Publicly available software, standard packages, and algorithms were used for the analysis, including Cell Ranger (v6.1.2) (RRID:SCR_017344), Seurat and built-in tools (v4.3.0) (RRID:SCR_016341), and UCSF ChimeraX (RRID:SCR_015872, v1.6). No custom code or algorithms were used or generated.

## Acknowledgements

We acknowledge the use of the Auditory and Vestibular Technology (AVT) Core of Translational Hearing Research Center at Creighton University for high-resolution confocal imaging and library preparation (10x Genomics), and the University of Nebraska DNA Sequencing Core Facility for scRNA-seq. The AVT core receives partial support from NIH grant 1P20GM139762-01 from NIGMS. The University of Nebraska DNA Sequencing Core receives partial support from the NCRR (RR018788). Scanning electron microscope was acquired and wholly funded by Nebraska EPSCoR award (Creighton-Department of Chemistry & Biochemistry). This research also utilized the computational resources of the NIH HPC Biowulf cluster (http://hpc.nih.gov) supported by the NIH Intramural research program. Funding National Institutes of Health grant IRP funds Z01-DC000002 from NIDCD (BK and AT). National Institutes of Health grant R01 DC016807 from NIDCD (DH).

## Additional information

### Funding

| Funder | Grant reference number | Author |
| --- | --- | --- |
| National Institutes of Health | R01 DC016807 | David Z He |
| National Institutes of Health | Z01-DC000002 | Bechara Kachar Amirrasoul Tavakoli |

The funders had no role in study design, data collection, and interpretation, or the decision to submit the work for publication.

### Author contributions

Zhenhang Xu, Data curation, Formal analysis, Validation, Investigation, Visualization, Methodology, Writing – review and editing; Amirrasoul Tavakoli, Conceptualization, Data curation, Formal analysis, Investigation, Visualization, Methodology, Writing – review and editing; Samadhi Kulasooriya, Validation, Investigation, Visualization, Methodology; Huizhan Liu, Validation, Investigation, Methodology; Shu Tu, Yi Li, Investigation, Methodology, Writing – review and editing; Celia Bloom, Visualization, Methodology, Writing – review and editing; Tirone D Johnson, Validation, Investigation, Methodology,

Writing – review and editing; Jian Zuo, Investigation, Visualization, Writing – review and editing; Litao Tao, Supervision, Visualization, Methodology, Writing – review and editing; Bechara Kachar, Conceptualization, Formal analysis, Supervision, Funding acquisition, Investigation, Visualization, Methodology, Project administration, Writing – review and editing; David Z He, Conceptualization, Formal analysis, Supervision, Funding acquisition, Validation, Investigation, Visualization, Methodology, Writing – original draft, Project administration, Writing – review and editing

### Author ORCIDs
Zhenhang Xu ⬤ https://orcid.org/0000-0001-9527-5207
Amirrasoul Tavakoli ⬤ https://orcid.org/0000-0002-7991-9858
Samadhi Kulasooriya ⬤ https://orcid.org/0009-0008-3553-4121
Huizhan Liu ⬤ https://orcid.org/0000-0003-2400-3284
Shu Tu ⬤ https://orcid.org/0000-0003-3692-6617
Yi Li ⬤ https://orcid.org/0000-0002-9986-6546
Tirone D Johnson ⬤ https://orcid.org/0000-0002-9279-4965
Jian Zuo ⬤ https://orcid.org/0009-0007-6278-6335
Litao Tao ⬤ https://orcid.org/0000-0002-9801-6515
Bechara Kachar ⬤ https://orcid.org/0000-0002-2803-8700
David Z He ⬤ https://orcid.org/0000-0001-8383-9634

### Ethics
This study was performed in strict accordance with the Guide for the Care and Use of Laboratory of the National Institutes of Health. The animal usage and care were approved by the Institutional Animal Care and Use Committees of Creighton University (Protocol #1046.3) and NIDCD (NIDCD ACUC Protocol #1215). Animals were euthanized for tissue collection and in vitro experiments using the methods acceptable by the American Veterinary Medical Association.

Reviewer #1 (Public review): https://doi.org/10.7554/eLife.108071.3.sa1
Reviewer #2 (Public review): https://doi.org/10.7554/eLife.108071.3.sa2
Author response https://doi.org/10.7554/eLife.108071.3.sa3

---

## Additional files

### Supplementary files
Supplementary file 1. Transcriptomes of four HC types.

Supplementary file 2. Genes/gene products candidates associated with axonemal components in vestibular kinocilia derived from the 96 nm repeat structures of the human respiratory axoneme (PDB: 8J07). Missing genes in mouse vestibular hair cells are in red.

Supplementary file 3. Microtubule inner proteins and microtubule-associated proteins absent from mouse HC scRNA-seq data.

MDAR checklist

### Data availability
Raw scRNA-seq datasets of adult cochlear and vestibular hair cells have been deposited to GEO (accession number GSE283534).

The following dataset was generated:

| Author(s) | Year | Dataset title | Dataset URL | Database and Identifier |
|---|---|---|---|---|
| Xu Z, Kulasooriya S, Liu H, He DZ | 2025 | scRNA-seq transcriptomic profiles of cochlear and vestibular hair cells from adult mice [10_weeks_CBAJ] | https://www.ncbi.nlm.nih.gov/geo/query/acc.cgi?acc=GSE283534 | NCBI Gene Expression Omnibus, GSE283534 |

The following previously published datasets were used:

| Author(s) | Year | Dataset title | Dataset URL | Database and Identifier |
|---|---|---|---|---|
| Burns JC, Kelly MC, Hoa M, Morell RJ, Kelley MW | 2015 | Single-cell RNA-Seq resolves cellular complexity in sensory organs from the neonatal inner ear | https://www.ncbi.nlm.nih.gov/geo/query/acc.cgi?acc=GSE71982 | NCBI Gene Expression Omnibus, GSE71982 |
| Barta CL, Liu H, Chen L, Li Y, Giffen KP, Kramer KL, Beisel KW, He DZ | 2017 | RNA-sequencing of Adult Zebrafish Inner Ear Hair Cells | https://www.ncbi.nlm.nih.gov/geo/query/acc.cgi?acc=GSE101693 | NCBI Gene Expression Omnibus, GSE101693 |
| Taha JA, Alan CG | 2024 | Single-cell transcriptomic analysis reveals increased regeneration in diseased human inner ears | https://www.ncbi.nlm.nih.gov/geo/query/acc.cgi?acc=GSE207817 | NCBI Gene Expression Omnibus, GSE207817 |
| Jen H-I | 2018 | Transcriptomic and epigenetic regulation of hair cell regeneration in the mouse utricle and its potentiation by Atoh1 | https://www.ncbi.nlm.nih.gov/geo/query/acc.cgi?acc=GSE121610 | NCBI Gene Expression Omnibus, GSE121610 |

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
