## [Editor Report · eLife Assessment]

Using single-cell transcriptomic data from mouse inner ear hair cells, the authors compare for the first time gene expression across the four recognized hair cell types in adults, generating information **fundamental** to understanding hair cell relationships between the ancient vestibular compartment and the more recent cochlea. Among observed differences, **compelling** evidence is provided for the expression in vestibular hair cells but not cochlear hair cells of certain ciliary motility-related genes, suggesting that the kinocilium of vestibular hair cells may function as an active force generator to increase sensitivity.

---

## [Referee Report · Reviewer #1 (Public review)]

Summary

From transcriptomic comparisons of adult mouse cochlear and vestibular hair cells, Xu et al. provide a broad and well-organized overview of differences across 4 established hair cell types (2 cochlear and 2 vestibular). They go on to demonstrate the power of such analyses to provide functional insights by focusing on the differentiated expression of ciliary genes, building to the hypothesis that kinociliary motility occurs in adult vestibular hair cells.

Background

Cilia are prominent in sensory receptors, including vertebrate photoreceptors, olfactory neurons and mechanosensitive hair cells of the inner ear and lateral line. Cilia can be motile or nonmotile depending on their axonemal structure: motile cilia require dynein and the inner 2 singlet microtubules of the 9+2 array. Primary cilia, present early in development, are considered to have sensory functions and to be nonmotile (Mill et al., Nature Rev Gen 2023).

In hair cells, the kinocilium anchors and polarizes the mechanosensitive hair bundle of specialized microvilli. The kinocilium matures from the primary cilium of a newborn hair cell; behind it the bundle of mechanosensory microvilli rises in a descending staircase of rows. During maturation of the mammalian cochlea, all hair cells lose the kinocilium, though not the associated basal body. The consensus for many years has been that most vertebrate kinocilia, and especially mammalian kinocilia, are nonmotile, based largely on the lack of spontaneous motility in excised mammalian vestibular organs, but also on the impression that the rare examples of spontaneous beating motility even in non-mammalian hair cells are associated with deterioration of the preparation (Rüsch & Thurm 1990).

Strengths

In comparing RNA expression across the 4 major types of mouse hair cells - 2 cochlear and 2 vestibular - Xu et al. provide rich data sets for exploration of structure-function differences between these highly specialized cell types. The revised paper significantly improves the organization, interpretation and readability of the presentation of overall findings. smFISH and immuno-staining back up key RNA data, and comparisons are made with published data.

The ciliary motility focus of the rest of the paper is creative and highly interesting. The authors curated the ciliary genes into types associated with different aspects of beating motility, and also investigated the expression of genes typical of primary cilia, which are considered to have sensory and cell signaling functions and to be nonmotile. Their data justify suggesting a role for kinociliary motility (or force generation) in adult mammalian vestibular hair cells, in opposition to a long-held assumption. The results should stimulate investigation of the implications for mechanosensitivity.

Weaknesses

Data

Functional data on kinocilia motility: The technical difficulty in making such measurements in small mouse hair bundles led the authors to work with bullfrog crista bundles. Though not extensively studied here, the ciliary motility shown is convincing. Mouse hair bundle motions are also shown but the evidence connecting the data to kinociliary motion are more suggestive than convincing. But the authors are not dogmatic about these data, and it is reasonable to show them.

Interpretation

The authors take the view that kinociliary motility is likely to be normally present but is rare in their observations because conditions are not right. But while others have described some (rare) kinociliary motility in fish organs (Rusch & Thurm 1990), they interpreted its occurrence as a sign of pathology. Indeed, in this paper, it is not clear what role kinociliary motility would play in mature hair bundles. The authors have added a discussion of this question in the revision.

An underlying rationale for the hypothesis that ciliary motility manifests in mammalian vestibular hair cells seems to rest on the presence of the necessary mRNA and its contrasting absence in cochlear hair cells. Another way to look at this difference could be that evolution acted on cochlear hair cells to shed kinocilia as one of many changes to improve mechanosensitivity at much higher sound frequencies. In vestibular hair cells, kinociliary motion might be useful to enhance mechanostimulation in the developing vestibule (as suggested in this revision) and not so active in maturity. Nevertheless, with their scholarly analysis of the expression of ciliary genes, the authors make a significant argument for further investigation of when and why hair cell kinocilia show active motility.

---

## [Referee Report · Reviewer #2 (Public review)]

Summary:

In this study the authors compared the transcriptomes of the various different types of hair cells contained in the sensory epithelia of the cochlea and vestibular organs of the mouse inner ear. The analysis of their transcriptomic data lead to novel insights into the potential function of the kinocilium.

Strengths:

The novel findings for the kinocilium gene expression along with the demonstration that some kinocilia demonstrate rhythmic beating as would be seen for known motile cilia is fascinating. It is possible that perhaps the kinocilium known to play a very important role in the orientation of the stereocilia, may have a gene expression pattern that is more like a primary cilium early in development and later in mature hair cells more like a motile cilium. Since the kinocilium is retained in vestibular hair cells it makes sense that it is playing a different role in these mature cells than its role in the cochlea.

Another major strength of this study which cannot be overstated is that for the transcriptome analysis they are using mature mice. To date there is a lot of data from many labs for embryonic and neonatal hair cells but very little transcriptomic data on the mature hair cells. They do a nice job in presenting the differences in marker gene expression between the 4 hair cell types. This information is very useful to those labs studying regeneration or generation of hair cells from ES cell cultures. One of the biggest questions these labs confront is what type of hair cell develop in these systems. The more markers available the better. These data will also allow researchers in the field to compare developing hair cells with mature hair cell to see what genes are only required during development and not in later functioning hair cells.

Comments on revision:

I am satisfied with the revision, the authors made an effort to incorporate the changes requested.

---

## [Author Response]

The following is the authors’ response to the original reviews.

**Public Reviews:**

**Reviewer #1 (Public review):**
Weaknesses:(1) Data:(a) The main weakness in the data is the lack of functional and anatomical data from mouse hair bundles. While the authors compensate in part for this difficulty with bullfrog crista bundles, those data are also fragmentary - one TEM and 2 exemplar videos. Much of the novelty of the EM depends on the different appearance of stretches of a single kinocilium - can we be sure of the absence of the central microtubule singlets at the ends?

Our single-cell RNA-seq findings show that genes related to motile cilia are specifically expressed in vestibular hair cells. This has not been demonstrated before. We have also provided supporting evidence using electrophysiology and imaging from bullfrogs and mice. Although no ultrastructural images of mouse vestibular kinocilia were provided in our study, transmission electron micrograph of mouse vestibular kinocilia has been published (O’Donnell and Zheng, 2022). The mouse vestibular kinocilia have a “9+2” microtubule configuration with nine doublet microtubules surrounding two central singlet microtubules. This finding contrasts with a previous study, which demonstrated that the vestibular kinocilia from guinea pigs lack central singlet microtubules and inner dynein arms, whereas outer dynein arms and radial spokes are present (Kikuchi et al., 1989). The central pair of microtubules is absent at the end of the bullfrog saccular kinocilium (Fig. 7A). We would like to point out that the dual identity of primary and motile cilia is not just based on the TEM images. The kinocilium has long been considered a specialized cilium, and its role as a primary cilium during development has been demonstrated before (Moon et al., 2020; Shi et al., 2022).

In most motile cilia, the central pair complex (CPC) does not originate directly from the basal body; instead, it begins a short distance above the transition zone, a feature that already illustrates variation in CPC assembly across systems (Lechtreck et al., 2013). The CPC can also show variation in its spatial extent: for example, in mammalian sperm axonemes, it can terminate before reaching the distal end of the axoneme (Fawcett and Ito, 1965). In addition, CPC orientation differs across organisms: in metazoans and *Trypanosoma*, the CPC is fixed relative to the outer doublets, whereas in *Chlamydomonas* and ciliates it twists within the axoneme (Lechtreck et al., 2013). Such variation has been described in multiple motile cilia and flagella and is therefore not unique to vestibular kinocilia. What appears more unusual in our data is the organization at the distal tip, where a distinct distal head is present, similar to cilia tip morphologies recently described in human islet cells (Polino et al., 2023). Although this feature is intriguing, we interpret it primarily as a structural signature rather than as evidence for a specialized motile adaptation, and we have moderated our interpretation accordingly in the revision.

(b) While it was a good idea to compare ciliary motility expression in published P2 datasets for mouse cochlear and vestibular hair cells for comparison with the authors' adult hair cell data, the presentation is too superficial to assess (Figure 6C-E; text from line 336) - it is hard to see the basis for concluding that motility genes are specifically lower in P2 cochlear hair cells than vestibular hair cells. Visually, it is striking that CHCs have much darker bands for about 10 motility-related genes.

While these genes (e.g., *Dynll1*, *Dynll2*, *Dynlrb1*, *Cetn2*, and *Mdh1*) appear more highly expressed in P2 cochlear hair cells, they are not uniquely associated with the axoneme. For example, *Dynll1/2* and *Dynlrb1* are components of the cytoplasmic dynein-1 complex (Pfister et al., 2006), Cetn2 has multiple basic cellular functions beyond cilia (e.g., centrosome organization, DNA repair), and *Mdh1* encodes a cytosolic malate dehydrogenase involved in central metabolic pathways such as the citric acid cycle and malate–aspartate shuttle. This contrasts with axonemal dyneins, which are uniquely required for cilia motility. To avoid ambiguity, we have marked such cytoplasmic or multifunctional genes with red asterisks in both Fig. 5G and Fig. 6D in the revised manuscript.

Our comparison showed that key genes for motile machinery are not detected in cochlear hair cells. For example, *Dnah6* and *Dnah5* are not expressed in the P2 cochlear hair cells. *Dnah6* and *Dnah5* encode axonemal dynein and are part of inner and outer dynein arms. Importantly, we did not detect the expression of CCDC39 and CCDC40 in kinocilia of P2 cochlear hair cells. Furthermore, axonemal CCDC39 and CCDC40, the molecular rulers that organize the axonemal structure in the 96-nm repeating interactome were not detected in cochlear hair cells. We have revised the text to emphasize key differences.

(2) Interpretation:The authors take the view that kinociliary motility is likely to be normally present but is rare in their observations because the conditions are not right. But while others have described some (rare) kinociliary motility in fish organs (Rusch & Thurm 1990), they interpreted its occurrence as a sign of pathology. Indeed, in this paper, it is not clear, or even discussed, how kinociliary motility would help with mechanosensitivity in mature hair bundles. Rather, the presence of an autonomous rhythm would actively interfere with generating temporally faithful representations of the head motions that drive vestibular hair cells.

Spontaneous flagella-like rhythmic beating of kinocilia in vestibular HCs in frogs and eels (Flock et al., 1977; Rüsch and Thurm, 1990) and in zebrafish early otic vesicle (Stooke-Vaughan et al., 2012; Wu et al., 2011) has been reported previously. Based on Rüsch and Thurm (1990), spontaneous kinocilia motility occurred under non-physiological conditions and was interpreted as a sign of cellular deterioration rather than a normal feature. We speculate that deterioration under non-physiological conditions may lead to the disruption of lateral links between the kinocilium and the stereociliary bundle, effectively unloading the kinocilium and allowing it to move more freely. Additionally, fluctuations in intracellular ATP levels may contribute, as ciliary motility is highly ATP-dependent; when ATP is depleted, beating ceases. Similar phenomena have been documented in respiratory epithelia, where ciliary activity can temporarily pause. Nevertheless, the fact that kinocilia can exhibit spontaneous motility under these conditions indicates that they possess the motile machinery necessary for such beating. Irrespective of the condition, cilia without the molecular machinery required for motility will not be able to move.

We agree with the reviewer that, based on the present data, it is difficult to know the functional role of kinocilia and whether the presence of such autonomous rhythm would interfere with temporal fidelity. Spontaneous bundle motion, driven by the active process associated with mechanotransduction, was observed in bullfrog saccular hair cells (Benser et al., 1996; Martin et al., 2003). We have revised the discussion to clarify this important point of the reviewer. Specifically, we will emphasize that our observations of ciliary beating in the ex vivo conditions may not reflect its properties in the mature in vivo context, but rather a byproduct of motile machinery clearly present in the kinocilia. We speculate that this machinery in mature hair cells could operate in a more subtle mode—modulating the rigor state of dynein arms or related axonemal structures to influence kinociliary mechanics and, in turn, bundle stiffness in response to stimuli or signaling cues. Such a mechanism could either enhance sensitivity or introduce filtering properties, thereby contributing to the fine control of mechanosensory function without compromising temporal fidelity. Future studies using loss-of-function approach will be needed to reveal the unexplored role(s) of kinocilia for vestibular hair cells in vertebrates.

We note that spontaneous activity exits throughout nervous system. It allows the nervous system to maintain baseline activity and interpret signals. Retinal cells are spontaneously active even in the dark and spiral ganglion neurons also fire spontaneously. Spontaneous hair bundle motion driven by mechanotransduction-related mechanism has been observed in bullfrog saccular hair cells. So, it is unlikely that spontaneous kinocilia beating would interfere with generating temporally faithful representations.

Could kinociliary beating play other roles, possibly during development - for example, by interacting with forming accessory structures (but see Whitfield 2020) or by activating mechanosensitivity cell-autonomously, before mature stimulation mechanisms are in place? Then a latent capacity to beat in mature vestibular hair cells might be activated by stressful conditions, as speculated regarding persistent Piezo channels that are normally silent in mature cochlear hair cells but may reappear when TMC channel gating is broken (Beurg and Fettiplace 2017). While these are highly speculative thoughts, there is a need in the paper for more nuanced consideration of whether the observed motility is normal and what good it would do.

We thank the reviewer for these excellent suggestions. We agree that kinociliary motility could plausibly serve roles during development, for example by guiding hair bundle formation or by contributing to early mechanosensitivity and spontaneous neural activity before mature stimulation mechanisms are established. It is also possible that the motility machinery represents a latent capacity in mature vestibular hair cells that could be reactivated under stress or pathological conditions. We have revised the Discussion to address these possibilities and to provide a more nuanced consideration of whether the observed motility is normal and what potential functions it might serve.

**Reviewer #2 (Public review):**
Summary:In this study, the authors compared the transcriptomes of the various types of hair cells contained in the sensory epithelia of the cochlea and vestibular organs of the mouse inner ear. The analysis of their transcriptomic data led to novel insights into the potential function of the kinocilium.Strengths:The novel findings for the kinocilium gene expression, along with the demonstration that some kinocilia demonstrate rhythmic beating as would be seen for known motile cilia, are fascinating. It is possible that perhaps the kinocilium, known to play a very important role in the orientation of the stereocilia, may have a gene expression pattern that is more like a primary cilium early in development and later in mature hair cells, more like a motile cilium. Since the kinocilium is retained in vestibular hair cells, it makes sense that it is playing a different role in these mature cells than its role in the cochlea.Another major strength of this study, which cannot be overstated, is that for the transcriptome analysis, they are using mature mice. To date, there is a lot of data from many labs for embryonic and neonatal hair cells, but very little transcriptomic data on the mature hair cells. They do a nice job in presenting the differences in marker gene expression between the 4 hair cell types. This information is very useful to those labs studying regeneration or generation of hair cells from ES cell cultures. One of the biggest questions these labs confront is what type of hair cells develop in these systems. The more markers available, the better. These data will also allow researchers in the field to compare developing hair cells with mature hair cells to see what genes are only required during development and not in later functioning hair cells.

We would like to thank reviewer 2 for his/her comments and hope that the datasets provided in this manuscript will be a useful resource for researchers in the auditory and vestibular neuroscience community.

**Joint Recommendations for the authors:**
(1) Figure 1 - Explain how hair cell types are recognized after dissociation. Figure 1 will not be clear in this regard for non-aficionados. Some of the dissociated cells shown appear quite distorted and even unhealthy - e.g., the bottom right crista type II hair cell; the second from left crista type I hair cell; can you address why this doesn't matter for the purposes of this study?

HC types in Fig. 1C were identified based on their morphological features: Type I HCs are flask-shaped with a narrow neck while type II HCs are cylindrical and short. We have replaced those cells with new images. In our study, HCs were identified based on their marker genes. Although some HCs such as those shown in Fig. 3C were impossible to avoid during preparation of single cells for library (most people did not examine their morphology), quality of mRNA and sequencing was high, better than those datasets published in previous studies.

(2) Line 98 - Explain accessory cells (as opposed to supporting cells).

We changed accessory cells to other cell types.

(3) Line 246 - The primary cilium is...

Changed.

(4) Figure 6D - The scale bar is missing. Please use arrows to point to the genes you call out in the text. Also, the genes called out in the text as differently expressed (line 342) are quite faint bands in both cell types. It would be a service to the reader to point them out in the panel.

A scale bar has been added. We also marked those genes as suggested and edited the text accordingly.

(5) Figure 7 - mixes frog crista and mouse middle ear images with waveforms and FFTs from frog crista, mouse middle ear, and mouse crista. Related to these still images are 2 videos of frog kinocilium beating (2 hair cells). The mouse images must be underwhelming, or we would have been shown those, yet they were considered adequate to analyze.

Yes, the spontaneous kinocilia motion of mouse crista HCs is very small. The peak motion is about 40 nm, which is very close to the resolution of our camera. That is why we used photodiode technique to detect its motion. Photodiode is more sensitive, and this technique allows us to observe dynamic response waveform.

(6) I recommend labeling each figure panel with the tissue of origin to avoid confusion.

Labeled as suggested.

(7) I suggest dropping the mouse middle ear data, as they are not directly adequate as a positive control (or no more so than the more beautiful frog data).

We keep the waveforms of middle ear cilia movement in Fig. 7. The main reason is that we would like to show the magnitude difference between airway cilia and kinocilia. The kinocilia movement was at least an order of magnitude less than the movement of airway cilia. This has led to our effort to generate a model to predict the 96-nm modular repeat and explain why kinocilia movement in mice is much smaller than airway cilia and bullfrog kinocilia.

(8) Focus on the hair bundle motions:(a) Show the waveforms for the frog crista hair cells and their FFTs.

These images were captured many years ago using camera. The kinocilia motion is between 5 and 10 Hz. We did not present any waveforms of kinocilia motion since we no longer have access to bullfrogs. However, although we did not present response waveforms, the videos are very powerful for visualization of kinocilia beat of bullfrog saccular HCs.

(b) Find some way to show us how you measured the mouse hair bundle beating.

Photodiode technique was used to measure spontaneous kinocilia motion in mice. More details are now included in the text.

(c) Does EGTA break links between kinocilium and stereocilia? (Could that contribute to the higher beat frequency?) Just applying the same treatment and viewing from above could clarify whether kinocilia dissociate from stereocilia rows. This would likely be more straightforward with an otolith organ.

All these links (tip links, side links) are vulnerable to Ca concentration and Ca-free medium is often used to break these links as shown in many previous studies. Breaking the kinocilia links leads to reduced load to the kinocilia, which may result in larger motion of the kinocilia. The frequency is inherent to motile machinery and subject to temperature and intracellular ATP concentration. When facing upward, the hair bundles in otolith organ do not have a good contrast against HCs in the background. This makes measurement of their motion difficult, especially when the motion is small and random and can’t be averaged to improve signal to noise ratio. Besides, unlike cochlear HCs whose hair bundles are short and can easily be oriented in parallel with light path, the long hair bundle of vestibular HCs is more difficult to orient and image. For these reasons, we chose to use crista hair bundles for our measurements since they can be oriented in perpendicular to the light path without interference from background HCs. The lateral motion of the entire bundle is also relatively easy to measure in this preparation.

(6) Is there no reason to cite McInturff et al. (2018), given that they compared type I and II VHC transcriptomes at P12 and P100? This database is also available on gEAR.

Their studies are now cited. We also compared their datasets with ours.

(7) Line 374 - Eatock et al., 1998 citation does not work for this purpose. Eatock & Songer (2011) would be better, or Li, Xue, Peterson (2008): mouse utricle anatomy; significant discussion of relative heights of kinocilia and tallest stereocilia.

Changed and cited.

(8) In Figure 3, 2 of the 18 panels in B are missing labels.

The bar, applied to all panels, was there at the bottom of Fig. 3B. The bar is bigger and more visible in the revision.

(9) Line 187 should "Sppl1" be Spp1?

Corrected.

(10) Define BBSome on line 244.

Added.

(11) Looking at Figure 5, it seems that all the motile genes are expressed in the vestibular hair cells and not the cochlear hair cells. It is surprising that there are any cilia-related genes expressed in these adult cochlear hair cells, given that they do not retain their cilia into adulthood. Could the authors make a comment on this finding in the discussion? Also, are there any ciliopathies that show a vestibular defect but normal hearing in mice or humans? Have you compared the cilia-related gene expression in neonatal/embryonic vestibular hair cells to your dataset?

There are many kinocilia related genes still expressing adult cochlear HCs. It is not surprising to see many kinocilia related genes in cochlear HCs. Most of these genes are related to primary cilia structure including the basal body and transporters in cilia. The basal body is still present in cochlear HCs. Many other primary cilia-related proteins are also expressed in soma, especially those related to signal transduction, microtubule cytoskeleton, actin cytoskeleton, vesicle transport, metabolic enzyme, protein folding, translation, nuclear transport, ubiquitination, RNA binding, mitochondrial proteins and transcription factors. Of course, some of them are vestigial. We added discussion of this in the text. Comparison between neonatal cochlear and vestibular was presented in Fig. 6D. We compared those genes related to the axonemal repeat (96 nm repeat complex). Due to quality of mRNA, the total genes and genes related to kinocilia detected in previous developmental studies were much less than our datasets. While we detected 112 out of 128 genes related to axonemal repeat, only 90 genes were detected in previous studies (Burns et al., 2015; McInturff et al., 2018). Therefore, we only compared neonatal cochlear and vestibular HCs using their datasets. As far as we know, no ciliopathies with vestibular defects but normal hearing have been reported in mice or humans. But we plan to use a *Ccdc39* mutant mouse model to examine how loss of function of a key motile cilia signature gene would affect kinocilia motility and vestibular function.

(12) How is "expression level" in the violin plots being calculated? Is this a measure of read count? The normalization is cursorily explained in the methods. Is this value comparable across genes? Did the authors switch to z-score by Figure 6?

We dissected the auditory and vestibular sensory epithelia from the same groups of mice and prepared libraries and sequenced them at the same time. All parameters are the same. The violin Plots are based on values presented in Supplementary Table 1. Each dot in the plot reflects an aggregated number of reads across all cells for each gene. They are all normalized across different HC types and biological repeats. The details for normalization are now provided.

(13) The authors comment on the 16/128 motile cilia axonemal repeat genes that are not expressed in the vestibular hair cells. Listing these somewhere may be helpful to the readers.

We thank the reviewer for this helpful suggestion. Most of the 128 motile cilia axonemal repeat genes were listed in Figs 8C and S5, along with known loss-of-function mutations and ciliopathy associations identified in human diseases or observed in animal models. To improve clarity, we have now included Table S2, which provides the complete list of all 128 motile cilia axonemal repeat genes, including those not expressed in vestibular HCs.

(14) Figure 5D needs some refinement. While the authors used databases, including CiliaCarta, SYSCILIA gold standard, and CilioGenics, to identify the primary cilia-related genes, they have included many genes that are not highly specific to primary cilia function (e.g., HSP90, HSPA8, DNAJA4, GNAS...). Perhaps the authors would be able to do a better job of specifically querying primary cilia function by using genes that are common to these three databases.

We presented comparison and analysis based on three major cilia databases, which are generated from proteomics of cilia from different tissues/organisms. In addition, we have provided more comprehensive list of primary cilia-related genes in Fig. S2. While majority of cilia-related genes/proteins are highly conserved, some genes/proteins are tissue-/organism-specific. Majority of the genes presented in Fig. 5D of our manuscript are shared among all three databases. The cilium is a complex structure, composed of proteins for microtubule cytoskeleton, actin cytoskeleton, vesicle transport, metabolic enzyme, signaling, and protein folding. It also contains proteins for translation, nuclear transport, ubiquitination, RNA binding as well as mitochondrial proteins and transcription factors (https://ciliogenics.com/?page=Home). Proteins such as HSP90 and HSPA8 are important for protein folding. HSPA8 also functions as an ATPase in the disassembly of clathrin-coated vesicles during transport of membrane components through the cell. GNAS is part of a G protein complex that transmits signals. DNAJA4 is one of the high-confidence cilia proteins (mean score of 1.26, expression rank is 938). These proteins are detected in cilia according to CilioGenics (https://ciliogenics.com/?page=Home). These proteins are not highly specific to cilia and are expressed in soma as well. Most of these proteins for signaling such as WNT (Supplementary Fig. 2) are detected in both cilia and soma.

(15) The authors state, "Furthermore, we observed robust spontaneous kinocilia motility in bullfrog crista HCs and small spontaneous bundle motion in mouse crista HCs." This statement should be moderated by acknowledging that this motility was observed in only some cells. The authors favor the hypothesis that the lack of motility in some crista HCs is due to depolarization or damage to the sample. The authors should also acknowledge the possibility that there may be cell-to-cell variability in the motility of the kinocilia.

We address these issues in public review section. We modified the statement as suggested.

(16) The first few pages of the Results section include many lists of genes. Readability may be improved if this is curtailed modestly.

Changed as suggested. We removed comparison among different types of HCs and replotted Fig. 2B. This has reduced the number of genes mentioned in the text.